# Dynamic Regret Bounds for Online Omniprediction with Long Term Constraints

## Abstract

We present an algorithm guaranteeing dynamic regret bounds for online omniprediction with long term constraints. The goal in this recently introduced problem is for a learner to generate a sequence of predictions which are broadcast to a collection of downstream decision makers. Each decision maker has their own utility function, as well as a vector of constraint functions, each mapping their actions and an adversarially selected state to reward or constraint violation terms. The downstream decision makers select actions "as if" the state predictions are correct, and the goal of the learner is to produce predictions such that all downstream decision makers choose actions that give them worst-case utility guarantees while minimizing worst-case constraint violation. Within this framework, we give the first algorithm that obtains simultaneous *dynamic regret* guarantees for all of the agents — where regret for each agent is measured against a potentially changing sequence of actions across rounds of interaction, while also ensuring vanishing constraint violation for each agent. Our results do not require the agents themselves to maintain any state — they only solve one-round constrained optimization problems defined by the prediction made at that round.

## 1 Introduction

In the problem of learning with long term constraints, there is a decision maker with an action space $\mathcal{A}$ and an adversary with an outcome space $\mathcal{Y}$. The learner has a utility function $u : \mathcal{A} \times \mathcal{Y} \to [0, 1]$, and a vector-valued constraint function $c : \mathcal{A} \times \mathcal{Y} \to [-1, 1]^d$. In rounds $t = 1, \ldots, T$, a *learner* chooses actions $a_t \in \mathcal{A}$ and an *adversary* chooses outcomes $y_t \in \mathcal{Y}$. The learner then obtains utility $u(a_t, y_t)$, and suffers a marginal constraint increment $c(a_t, y_t)$. The goal of the learner is to satisfy all of the constraints marginally (up to a vanishing regret term) — i.e. to guarantee that for all sequences of outcomes:

$$\mathrm{CCV}(1 : T) \doteq \max_j \sum\nolimits_{t=1}^{T} c_j(a_t, y_t) \le o(T),$$

while simultaneously guaranteeing some notion of *regret* to the best action in some benchmark class. It has been known since Mannor et al. (2009) that in adversarial settings it is not possible to compete against the best fixed action in hindsight that satisfies the constraints *marginally*. Instead, the standard benchmark in this literature is the set of actions that in hindsight satisfy the realized constraints *every round*:

$$\mathcal{A}_{1:T}^{\mathbf{c}} = \{a \in \mathcal{A} : c_j(a, y_t) \le 0 \text{ for every } t \in [T] \text{ and } j \in [J]\}.$$

The corresponding standard goal in this literature (see e.g. Sun et al. (2017); Castiglioni et al. (2022); Qiu et al. (2023); Sinha & Vaze (2024)) is to minimize external regret with respect to this benchmark:

$$\mathrm{Reg}_{\mathrm{ext}}(1 : T) \doteq \max_{a \in \mathcal{A}_{1:T}^{\mathbf{c}}} \sum\nolimits_{t=1}^{T} (u(a, y_t) - u(a_t, y_t)) \le o(T).$$

A more ambitious goal, studied by a sub-thread of this literature, is to compete with a *changing benchmark sequence of actions*, so long as the benchmark sequence does not change too quickly (Chen et al., 2017; 2018; Chen & Giannakis, 2018; Cao & Liu, 2019; Vaze, 2022; Liu et al., 2022a; Lekeufack & Jordan, 2024). This is called a *dynamic regret* benchmark. For continuous action spaces, there are a variety of ways to measure "change", but we state here a version for discrete

categorical action spaces, which is the focus of our paper. First we define a richer benchmark that allows for *changing* sequences of benchmark actions that satisfy the constraints at each round.

$$\mathcal{A}_{1:T}^{\mathrm{dyn}} = \{\vec{a} \in \mathcal{A}^T : c_j(\vec{a}_t, y_t) \leq 0 \text{ for every } t \in [T] \text{ and } j \in [J]\}.$$

For a given sequence of actions $\vec{a} \in \mathcal{A}_{1:T}^{\mathrm{dyn}}$, we write $\Delta(\vec{a}) = |\{t : \vec{a}_t \neq \vec{a}_{t+1}\}|$ for the number of times the action changes in the benchmark sequence. The goal is to obtain diminishing dynamic regret:

$$\max_{\vec{a} \in \mathcal{A}_{1:T}^{\mathrm{dyn}}} \sum\nolimits_{t=1}^{T} \left( u(\vec{a}_t, y_t) - u(a_t, y_t) \right) - \mathrm{Reg}_{\mathrm{ext-dyn}}(\vec{a}) \leq 0.$$

Here we want the "dynamic regret bound" $\mathrm{Reg}_{\mathrm{ext-dyn}}(\vec{a})$ to be $o(T)$ for all $\vec{a}$ such that $\Delta(\vec{a}) \leq o(T)$.

Dynamic regret bounds are a very natural objective to strive for. The reason that we study a sequential adversarial environment is that we expect the environment to change (in potentially unpredictable ways) over time. In such environments, we naturally expect the optimal decision to also change over time, and in the constrained optimization setup, we might worry that the static benchmark $\mathcal{A}_{1:T}^{\mathbf{c}}$ is empty, even if there are feasible actions at every time step. Hence, the literature on online learning with long term constraints studying external regret generally make a rather strong assumption regarding the existence of a *single action* that satisfies all of the constraints across *all rounds*. Dynamic regret bounds, on the other hand, overcome this concern for slowly changing environments, by allowing for changes within the benchmark sequence while only requiring *local feasibility* — that the action we compare to on each round satisfies the constraints of that specific round (and not across all rounds of the interaction).

While the literature on online learning with long term constraints generally couples the problem of predicting outcomes $y_t$ and choosing actions $a_t$ by focusing on algorithms for a *single decision maker*, Bechavod et al. (2025) recently introduced the *omniprediction* variant of this problem. Omniprediction (c.f. Gopalan et al. (2022)) is a framework that decouples prediction from decision making by modeling a "predict-as-a-service" setting. In this framework, a single, centralized forecaster produces predictions that can be simultaneously consumed by many downstream decision makers who have different utilities, while providing guarantees for all the decision makers.

Bechavod et al. (2025) introduced the problem of "omniprediction with long term constraints", which extends the omniprediction framework to the setting where downstream decision makers also have different constraint functions. This is motivated by many real-world systems. For example, consider an energy management system where predictions about electricity prices are broadcast. These predictions are consumed by diverse agents: residential users may defer appliance usage to low-cost periods, manufacturers may have strict production quotas and budget constraints. Bechavod et al. (2025) showed how to make predictions that simultaneously guarantee all such decision makers worst-case regret and constraint violation bounds with respect to the benchmark class $\mathcal{A}_{1:T}^{\mathbf{c}}$, but their algorithm does not extend to give dynamic regret bounds. In this work we show how to make predictions in a way that gives dynamic regret bounds (in fact stronger dynamic *swap* regret bounds) simultaneously for many downstream decision makers.

A detailed discussion of additional related work is deferred to Appendix A.

## 1.1 OUR RESULTS

**Better Subsequence Regret Bounds** Previous work on online omniprediction with long term constraints gave regret bounds that held not just marginally over the whole sequence, but simultaneously on an arbitrarily specified collection of *subsequences* of it. In principle dynamic regret bounds can be extracted from subsequence regret bounds like this, by taking the set of subsequences to be the set of all $\approx T^2$ contiguous intervals in $\{1, \ldots, T\}$. Unfortunately the bounds obtained by Bechavod et al. (2025) depend *linearly* on the number of specified subsequences, which does not yield nontrivial dynamic regret bounds. Our main contribution is a new algorithm giving regret and constraint violation guarantees on arbitrary subsequences, with a dependence on the number of subsequences scaling only *logarithmically*. Dynamic regret bounds fall out as a special case.

**Stronger Notions of Regret** In fact, since our subsequence regret bounds are stronger *swap regret* bounds, the dynamic regret bounds we obtain are stronger than those that have been previously

studied in the literature on learning with long term constraints: we give bounds on what we call *dynamic swap regret*. Our new benchmark allows each decision maker to compete with a sequence of actions that results from applying a *swap function* remapping the decision maker's realized actions to alternatives. We allow the swap function itself to change with time. The traditional notion of dynamic regret is the special case in which these swap functions are constant valued.

**Easier Implementation for Downstream Agents**   Finally, Bechavod et al. (2025) obtained their results by requiring downstream agents to map predictions to actions using an elimination-based algorithm, which required all agents to actively maintain state — the set of actions that had not yet violated any of their constraints. Our algorithm allows downstream decision makers to map predictions to actions in an entirely stateless way: they simply evaluate both their constraint function and objective function as if our predictions were correct, and take the action that solves the resulting one-round constrained optimization problem.

## 2    Model and Preliminaries

The model largely follows the framework for online omniprediction with long-term constraints introduced in Bechavod et al. (2025). Let $\mathcal{X}$ denote the feature space and $\mathcal{Y}$ denote the outcome/label space. Throughout, we consider $\mathcal{Y} = [0, 1]^d$. We consider a set of agents $\mathcal{N}$ with an arbitrary discrete action space $\mathcal{A}$. Each agent is equipped with a tuple $(u, c_1, \ldots, c_J)$, which includes a utility function $u : \mathcal{A} \times \mathcal{Y} \to [0, 1]$ and $J$ constraint functions $\{c_j : \mathcal{A} \times \mathcal{Y} \to [-1, 1]\}_{j \in [J]}$[1]. We also sometimes write the constraint functions as a single vector valued function $\mathbf{c} = (c_1, \ldots, c_J)$. Since agents are uniquely defined by their corresponding tuples, we treat agents and their tuples interchangeably. We make the following assumptions on the utility and constraint functions.

**Assumption 1.** *We assume that for every agent $(u, \mathbf{c}) \in \mathcal{N}$ and action $a \in \mathcal{A}$, the utility function $u(a, \cdot)$ and constraint functions $c_j(a, \cdot)$ are linear and Lipschitz-continuous in $y$. Specifically, for any $f \in \{u, c_1, \ldots, c_J\}$, we require $f(a, k_1 y_1 + k_2 y_2) = k_1 f(a, y_1) + k_2 f(a, y_2)$ for all $k_1, k_2 \in \mathbb{R}$ and $y_1, y_2 \in \mathcal{Y}$. Furthermore, we assume there exist universal constants $L_{\mathcal{U}}$ and $L_{\mathcal{C}}$ such that for all $y_1, y_2 \in \mathcal{Y}$: $|u(a, y_1) - u(a, y_2)| \leq L_{\mathcal{U}} \|y_1 - y_2\|_\infty$ and $|c_j(a, y_1) - c_j(a, y_2)| \leq L_{\mathcal{C}} \|y_1 - y_2\|_\infty$.*

While the assumptions on $u$ are standard in the omniprediction literature (Bechavod et al., 2025), we extend these assumptions to the constraint functions $\mathbf{c}$. This will enable the purely prediction-based decision rule we introduce later, where agents select actions that are predicted to be feasible without needing to track historical constraint violations as was required in Bechavod et al. (2025).

**Remark 1.** *For simplicity we assume that the utility functions and constraint functions are* linear *in $y$, but we can equally well handle the case in which the utility functions are* affine *in $y$, as we can augment the label space with an additional coordinate that takes constant value 1. This preserves the convexity of the label space and allows for arbitrary constant offsets in the utility/constraint of each action. Assuming linear/affine utility functions is only more general than the standard assumption that decision makers are* risk neutral *in the sense that in the face of randomness, decision makers act to maximize their expected utility. If $\mathcal{Y}$ represents the set of probability distributions over outcomes, any risk neutral decision maker has a linear utility function by linearity of expectation.*

We take the role of an online/sequential forecaster producing predictions that will be consumed by agents. We consider the following repeated interaction between a forecaster, agents, and an adversary. In every round $t \in [T]$:

(1) The adversary selects a feature vector $x_t \in \mathcal{X}$ and a distribution over outcomes $Y_t \in \Delta \mathcal{Y}$;

(2) The forecaster observes the feature $x_t$, produces a distribution over predictions $\pi_t \in \Delta \mathcal{Y}$, from which a prediction $p_t \in \mathcal{Y}$ is sampled;

(3) Each agent chooses an action $a_t$ as a function of the prediction $p_t$ and the history;

(4) The adversary reveals an outcome $y_t \sim Y_t$, and the agent obtains utility $u(a_t, y_t)$ and the constraint loss vector $\{c_j(a_t, y_t)\}_{j \in [J]}$.

---

[1]Sometimes online adversarial learning problems are described by an adversary choosing a different utility and/or constraint function at each step. This is equivalent to having a fixed state-dependent utility function/constraint functions, and having an adversary choose state.

**Remark 2.** *Our framework makes no assumptions about the relationship between the feature $x_t$ and the outcome distribution $Y_t$. Our algorithm is designed to provide guarantees whether $x_t$ is highly informative or completely uninformative about $Y_t$. Furthermore, if one has access to a class of predictive models that predict the outcomes using the features, our framework can be configured to incorporate them and produce predictions that perform at least as well as the best model in that class. We discuss this extension in Appendix F.*

We will focus on performance over a collection of subsequences $\mathcal{S}$, where each subsequence $S \in \mathcal{S}$ is a subset of $[T]$. These subsequences need not be fixed in advance but can be defined dynamically. A subsequence $S \in \mathcal{S}$ is generally characterized by an indicator function $h_S : [T] \times \mathcal{X} \to \{0,1\}$. For any round $t \in [T]$, the round is part of the subsequence $S$ if and only if $h_S(t, x_t) = 1$. This flexible definition allows subsequences to be based on the round index $t$, the feature $x_t$, or both.

Agents aim to maximize their cumulative utilities over every subsequence in $\mathcal{S}$: $\sum_{t \in S} u(a_t, y_t)$ while minimizing their cumulative constraint violation (CCV) over every subsequence in $\mathcal{S}$:

$$\text{CCV}(S) := \max_{j \in [J]} \sum_{t \in S} c_j(a_t, y_t) \leq o(|S|).$$

We treat utility maximization as an objective and cumulative constraint violation as a requirement: $\text{CCV}(S)$ must be sublinear in the length of the subsequence, $|S|$.

We measure performance against different benchmark classes. The fundamental building block for our benchmark classes is the set of actions that are feasible at a specific round $t \in [T]$ with a margin of $\lambda \geq 0$:

$$\mathcal{A}_t^{\mathbf{c},\lambda} = \{a \in \mathcal{A} : c_j(a, y_t) \leq -\lambda \text{ for every } j \in [J]\}.$$

The margin $\lambda$ parameterizes the difficulty of the benchmarks; a smaller $\lambda$ yields a larger and thus more competitive set of actions[2]. We note that this margin can be set independently for each subsequence $S \in \mathcal{S}$, and our final results will be achieved by choosing its value as a function of the subsequence length $|S|$.

**Remark 3.** *Bechavod et al. (2025) also considered the benchmark actions that satisfy the constraints in expectation, i.e., $\mathcal{A}_t^{\mathbb{E}[\mathbf{c}],\lambda} = \{a \in \mathcal{A} : \mathbb{E}_{y_t \sim Y_t}[c_j(a, y_t)] \leq -\lambda \text{ for every } j \in [J]\}$. This benchmark class can be more challenging to compete with than $\mathcal{A}_t^{\mathbf{c},\lambda}$, since a single bad outcome $c_j(a, y_t) > 0$ does not immediately disqualify an action. Consequently, Bechavod et al. (2025) proposed different algorithms to handle the two benchmark classes. In contrast, our approach is general enough to handle $\mathcal{A}_t^{\mathbb{E}[c],\lambda}$ just as it does $\mathcal{A}_t^{c,\lambda}$. For clarity, we focus on $\mathcal{A}_t^{c,\lambda}$ in the main text and defer the discussion of why our method applies to $\mathcal{A}_t^{\mathbb{E}[\mathbf{c}],\lambda}$ to Appendix E.*

In the literature on learning with long-term constraints, a standard benchmark is the set of actions that are feasible at every round. We generalize this to our multi-subsequence framework by requiring this condition to hold throughout a given subsequence $S \in \mathcal{S}$:

$$\mathcal{A}_S^{\mathbf{c},\lambda} = \cap_{t \in S} \mathcal{A}_t^{\mathbf{c},\lambda} = \{a \in \mathcal{A} : c_j(a, y_t) \leq -\lambda \text{ for every } t \in S \text{ and } j \in [J]\}.$$

Competing with the best fixed action from this class in hindsight leads to the notion of *constrained external regret*.

**Definition 1** (Constrained External Regret over Subsequence $S$)**.** *Fix an agent with a utility function $u : \mathcal{A} \times \mathcal{Y} \to [0,1]$ and a constraint function $\mathbf{c} : \mathcal{A} \times \mathcal{Y} \to [-1,1]^J$. Fix a subsequence $S \subseteq [T]$. For a sequence of actions $a_1, \ldots, a_T$ and outcomes $y_1, \ldots, y_T$, the agent's constrained external regret over the subsequence $S$ is:*

$$\text{Reg}_{\text{ext}}(u, \mathbf{c}, \lambda, S) = \max_{a \in \mathcal{A}_S^{\mathbf{c},\lambda}} \sum_{t \in S} (u(a, y_t) - u(a_t, y_t)).$$

We will also compete with a more demanding benchmark based on action modification rules. For the benchmark class $\mathcal{A}_S^{\mathbf{c},\lambda}$, an action modification rule is any function $\phi : \mathcal{A} \to \mathcal{A}_S^{\mathbf{c},\lambda}$ that consistently

---

[2]Assuming the existence of such a strongly feasible action is known in the literature on learning with long term constraints as Slater's condition (Neely & Yu, 2017; Chen et al., 2017; Cao & Liu, 2019; Yu & Neely, 2020; Castiglioni et al., 2022)). We later additionally give bounds that hold without making this assumption.

maps an agent's actions to alternatives within the benchmark class. Competing with the best such rule in hindsight leads to the notion of *constrained swap regret*. This is a stronger notion than constrained external regret, as constrained external regret can be viewed as a special case where the action modification rule is restricted to being a constant function.

**Definition 2** (Constrained Swap Regret over Subsequence $S$). *Fix an agent with a utility function $u : \mathcal{A} \times \mathcal{Y} \to [0,1]$ and a constraint function $\mathbf{c} : \mathcal{A} \times \mathcal{Y} \to [-1,1]^J$. Fix a subsequence $S \subseteq [T]$. For a sequence of actions $a_1, \ldots, a_T$ and outcomes $y_1, \ldots, y_T$, the agent's constrained swap regret over the subsequence $S$ is:*

$$\mathrm{Reg}_{\mathrm{swap}}(u, \mathbf{c}, \lambda, S) = \max_{\phi:\mathcal{A}\to\mathcal{A}_S^{\mathbf{c},\lambda}} \sum_{t\in S} \left(u(\phi(a_t), y_t) - u(a_t, y_t)\right).$$

When $\mathcal{S}$ is the set of all contiguous intervals, $\mathcal{S} = \{[t_1, t_2] : 1 \leq t_1 \leq t_2 \leq T\}$, we refer to the resulting instantiations of Definitions 1 and 2 as *constrained external adaptive regret* and *constrained swap adaptive regret*, respectively.

While adaptive regret provides a powerful guarantee over all contiguous intervals, a different but related goal is to measure performance against a dynamic benchmark path that changes over time, known as dynamic regret. We now introduce the external and swap versions of this guarantee under our framework with long-term constraints.

The benchmarks for *constrained external dynamic regret* are changing sequences of benchmark actions that satisfy the constraints (with a fixed margin of $\lambda$) at each round:

$$\vec{\mathcal{A}}_{1:T}^{\mathbf{c},\lambda} = \prod_{t=1}^{T} \mathcal{A}_t^{\mathbf{c},\lambda} = \{\vec{a} \in \mathcal{A}^T : c_j(\vec{a}_t, y_t) \leq -\lambda \text{ for every } t \in [T] \text{ and } j \in [J]\}.$$

The complexity of any such benchmark sequence $\vec{a} \in \vec{\mathcal{A}}_{1:T}^{\mathbf{c},\lambda}$ is measured by the number of times the action changes: $\Delta(\vec{a}) = |\{t \in [T-1] : \vec{a}_t \neq \vec{a}_{t+1}\}|$.

**Definition 3** (Constrained External Dynamic Regret). *Fix an agent with a utility function $u : \mathcal{A} \times \mathcal{Y} \to [0,1]$ and a constraint function $\mathbf{c} : \mathcal{A} \times \mathcal{Y} \to [-1,1]^J$. For a sequence of actions $a_1, \ldots, a_T$ and outcomes $y_1, \ldots, y_T$, the agent's constrained external dynamic regret against a benchmark sequence of actions $\vec{a} \in \mathcal{A}^T$ is:*

$$\mathrm{Reg}_{\mathrm{ext-dyn}}(u, \vec{a}) = \sum_{t=1}^{T} \left(u(\vec{a}_t, y_t) - u(a_t, y_t)\right).$$

*The goal is to guarantee that $\mathrm{Reg}_{\mathrm{ext-dyn}}(u, \vec{a})$ is $o(T)$ for any benchmark sequence $\vec{a}$ that is (1) dynamically feasible, $\vec{a} \in \vec{\mathcal{A}}_{1:T}^{\mathbf{c},\lambda}$, and (2) has a sublinear number of changes, $\Delta(\vec{a}) = o(T)$.*

Next, we define the more powerful swap-based counterpart. The benchmarks for *constrained swap dynamic regret* are changing sequences of action modification rules $\vec{\phi} \in (\mathcal{A}^{\mathcal{A}})^T$. The complexity of any such benchmark sequence is likewise measured by the number of times the action modification rule changes: $\Delta(\vec{\phi}) = |\{t \in [T-1] : \vec{\phi}_t \neq \vec{\phi}_{t+1}\}|$. A sequence with $\Delta(\vec{\phi})$ changes partitions the time horizon $[1, T]$ into $\Delta(\vec{\phi}) + 1$ contiguous intervals, on each of which the rule is fixed. We only compete with sequences where, for each interval of constancy, the fixed action modification rule maps to the set of actions that are feasible throughout that entire interval.

**Definition 4** (Constrained Swap Dynamic Regret). *Fix an agent with a utility function $u : \mathcal{A} \times \mathcal{Y} \to [0,1]$ and a constraint function $\mathbf{c} : \mathcal{A} \times \mathcal{Y} \to [-1,1]^J$. For a sequence of actions $a_1, \ldots, a_T$ and outcomes $y_1, \ldots, y_T$, the agent's constrained swap dynamic regret against a benchmark sequence of action modification rules $\vec{\phi} \in (\mathcal{A}^{\mathcal{A}})^T$ is:*

$$\mathrm{Reg}_{\mathrm{swap-dyn}}(u, \vec{\phi}) = \sum_{t=1}^{T} \left(u(\vec{\phi}_t(a_t), y_t) - u(a_t, y_t)\right).$$

*The goal is to guarantee that $\mathrm{Reg}_{\mathrm{swap-dyn}}(u, \vec{\phi})$ is $o(T)$ for any benchmark sequence $\vec{\phi}$ that satisfies two properties: (1) It is piecewise feasible. Let the sequence have change points that partition $[1, T]$ into intervals $I_0, \ldots, I_{\Delta(\vec{\phi})}$. The action modification rule on each interval $I_k$ is the constant rule $\psi_k : \mathcal{A} \to \mathcal{A}_{I_k}^{\mathbf{c},\lambda}$. (2) It has a sublinear number of changes, $\Delta(\vec{\phi}) = o(T)$.*

Any dynamic benchmark sequence (either $\vec{a}$ or $\vec{\phi}$) with $\Delta$ changes naturally partitions the time horizon $[1, T]$ into $\Delta + 1$ contiguous intervals based on its change points. Within each of these intervals, the benchmark is fixed. Since an adaptive regret guarantee ensures low regret over *all* possible intervals, it also ensures low regret over this specific partition. The total dynamic regret can therefore be bounded by summing the regret bounds over these $\Delta + 1$ intervals, hence a low adaptive regret bound implies a low dynamic regret bound.

We will first focus on the strictly feasible benchmark classes with a margin of $\lambda = \tilde{\Omega}(1/\sqrt{T})$, and obtain $\tilde{O}(\sqrt{T})$ regret and $\tilde{O}(\sqrt{T})$ cumulative constraint violation; We will then apply similar techniques to handle the nominally feasible benchmark classes with zero margin ($\lambda = 0$), and obtain $\tilde{O}(T^{2/3})$ regret and $\tilde{O}(T^{2/3})$ cumulative constraint violation.

We note that all benchmark classes discussed are agent-specific as they depend on agents' constraint functions. All guarantees we provide will be stated under the assumption that the corresponding benchmark class is non-empty.

## 3 Proposed Approach and Main Guarantees

### 3.1 A Purely Prediction-Based Decision Rule

We propose a stateless decision rule for the agents that is purely based on the forecaster's predictions. At each round $t \in [T]$, each agent acts as if the prediction $p_t$ were accurate, and chooses the action that offers the highest predicted utility among all actions predicted to be feasible. We say that agents play *constrained best responses* to the predictions.

**Definition 5** (Constrained Best Response). *Fix a utility $u : \mathcal{A} \times \mathcal{Y} \to [0, 1]$, $J$ constraints $\{c_j : \mathcal{A} \times \mathcal{Y} \to [-1, 1]\}_{j \in [J]}$, and a prediction $p \in \mathcal{Y}$. The constrained best response to $p$ according to $u$ and $\{c_j\}_{j \in [J]}$, denoted as $\mathrm{CBR}^{u,\mathbf{c}}(p)$, is the solution to the constrained optimization problem:*

$$\begin{aligned} \underset{a \in \mathcal{A}}{maximize} \quad & u(a, p_t) \\ subject\ to \quad & c_j(a, p_t) \le 0 \text{ for every } j \in [J] \end{aligned}$$

The agent can obtain $\mathrm{CBR}^{u,\mathbf{c}}(p_t)$ in two steps:

(1) The agent first discards actions that are infeasible according to the prediction.

For each constraint $j \in [J]$, the set of actions predicted to violate that constraint is denoted as:

$$\widehat{\mathcal{A}}_t^{c_j, \inf} = \{a \in \mathcal{A} : c_j(a, p_t) > 0\}.$$

The agent discards actions that are predicted to violate any of the $J$ constraints, i.e.,

$$\widehat{\mathcal{A}}_t^{\mathbf{c}, \inf} = \cup_{j \in [J]} \widehat{\mathcal{A}}_t^{c_j, \inf} = \{a \in \mathcal{A} : \exists j \in [J], c_j(a, p_t) > 0\}.$$

The retained actions that are predicted to be feasible are denoted as

$$\widehat{\mathcal{A}}_t^{\mathbf{c}, \text{fea}} = \{a \in \mathcal{A} : \forall j \in [J], c_j(a, p_t) \le 0\}.$$

(2) The agent then chooses an action from the retained action set $\widehat{\mathcal{A}}_t^{\mathbf{c}, \text{fea}}$ that maximizes the utility function according to the prediction, i.e.,

$$a_t = \arg\max_{a \in \widehat{\mathcal{A}}_t^{\mathbf{c}, \text{fea}}} u(a, p_t).$$

If none of the actions are predicted to be feasible, i.e., $\widehat{\mathcal{A}}_t^{\mathbf{c}, \text{fea}} = \emptyset$, the agent can choose any arbitrary action from $\mathcal{A}$. As we will formally prove, our predictions ensure this special case rarely occurs, and hence its influence on the cumulative constraint violation and regret is negligible.

## 3.2 Conditionally Unbiased Predictions

To ensure that our predictions are trustworthy so that treating them as the truth is a sensible choice for the agents, we need the predictions to be unbiased — not only marginally, but also conditionally on various subsequences. Notably, we define the following notions of conditional unbiasedness. The first is a "constrained" variant of decision calibration as defined in Noarov et al. (2023), which itself is a strengthening of an earlier notion of decision calibration due to Zhao et al. (2021)

**Definition 6** $((\mathcal{N}, \mathcal{S}, \alpha)$-Decision Calibration). *Let $\mathcal{S}$ be a collection of subsequences. Let $\mathcal{N}$ be a set of agents, where each agent is equipped with a utility function $u : \mathcal{A} \times \mathcal{Y} \to [0, 1]$ and $J$ constraint functions $\{c_j : \mathcal{A} \times \mathcal{Y} \to [-1, 1]\}_{j \in [J]}$. We say that a sequence of predictions $p_1, \ldots, p_T$ is $(\mathcal{N}, \mathcal{S}, \alpha)$-decision calibrated with respect to a sequence of outcomes $y_1, \ldots, y_T$ if for every $S \in \mathcal{S}$, $a \in \mathcal{A}$, and $(u, \mathbf{c}) \in \mathcal{N}$:*

$$\left\| \sum\nolimits_{t=1}^{T} \mathbb{1}\left[t \in S, \mathrm{CBR}^{u,\mathbf{c}}(p_t) = a\right](p_t - y_t) \right\|_{\infty} \leq \alpha(T^{u,\mathbf{c},S}(a)),$$

*where $T^{u,\mathbf{c},S}(a) = \sum_{t=1}^{T} \mathbb{1}\left[t \in S, \mathrm{CBR}^{u,\mathbf{c}}(p_t) = a\right]$.*

Decision calibration guarantees that forecasts are unbiased conditional on the decisions of the downstream decision makers. Infeasibility calibration, defined next, requires that the predictions be unbiased conditional on each action for each downstream decision maker being predicted to be infeasible.

**Definition 7** $((\mathcal{N}, \mathcal{S}, \beta)$-Infeasibility Calibration). *Let $\mathcal{S}$ be a collection of subsequences. Let $\mathcal{N}$ be a set of agents, where each agent is equipped with a utility function $u : \mathcal{A} \times \mathcal{Y} \to [0, 1]$ and $J$ constraint functions $\{c_j : \mathcal{A} \times \mathcal{Y} \to [-1, 1]\}_{j \in [J]}$. We say that a sequence of predictions $p_1, \ldots, p_T$ is $(\mathcal{N}, \mathcal{S}, \beta)$-infeasibility calibrated with respect to a sequence of outcomes $y_1, \ldots, y_T$ if for every $S \in \mathcal{S}$, $a \in \mathcal{A}$, $(u, \mathbf{c}) \in \mathcal{N}$, and $j \in [J]$:*

$$\left\| \sum\nolimits_{t=1}^{T} \mathbb{1}\left[t \in S, a \in \widehat{\mathcal{A}}_t^{c_j, \mathrm{inf}}\right](p_t - y_t) \right\|_{\infty} \leq \beta(T^{c_j, S, \mathrm{inf}}(a)),$$

*where $T^{c_j, S, \mathrm{inf}}(a) = \sum_{t=1}^{T} \mathbb{1}\left[t \in S, a \in \widehat{\mathcal{A}}_t^{c_j, \mathrm{inf}}\right]$.*

**Assumption 2.** *We assume that $\alpha, \beta : \mathbb{R} \to \mathbb{R}$ are concave functions. This will be the case in all the bounds we give; in general, this condition holds for any sublinear error bound $T^r$ for $r < 1$.*

In the sections that follow, we will show how these two unbiasedness constraints lead to bounds on the cumulative constraint violation and regret for all downstream decision makers.

We now address the algorithmic challenge of producing predictions that are decision calibrated and infeasibility calibrated. Our approach builds upon the Unbiased-Prediction algorithm from Noarov et al. (2023), which makes conditionally unbiased predictions in the online setting. We provide the pseudocode in Algorithm 1; for the formal guarantees and analysis, we refer readers to Appendix D and the original work.

At a high level, the Unbiased-Prediction algorithm provides a general method for making predictions $p_t$ that are unbiased with respect to a collection of "events" $\mathcal{E}$. Formally, an event $E \in \mathcal{E}$ is a function that maps the feature $x_t$ and the prediction $p_t$ to a value $E(x_t, p_t)$ in $[0, 1]$. The algorithm's goal is to ensure that for every event $E \in \mathcal{E}$, the cumulative prediction bias conditioning on that event is small, i.e., $\left\| \sum_{t=1}^{T} E(x_t, p_t)(p_t - y_t) \right\|_{\infty}$ is sublinear in $T$.

To achieve this, the algorithm reduces the problem to a zero-sum game between a learner and an adversary. At each round, it computes an approximate minimax equilibrium of this game: the learner chooses a distribution over predictions $\psi_t$ to minimize the weighted sum of conditional biases (emphasizing events with large accumulated bias), while the adversary tries to maximize it.

We ensure our two calibration properties by instantiating this algorithm with a specific collection of events. This collection is the union of all events required by Definition 6 and Definition 7, namely all events of the form $\mathbb{1}\left[t \in S, \mathrm{CBR}^{u,\mathbf{c}}(p_t) = a\right]$ and $\mathbb{1}\left[t \in S, a \in \widehat{\mathcal{A}}_t^{c_j, \mathrm{inf}}\right]$. We refer to this specific instantiation of Unbiased-Prediction as Decision-Infeasibility-Calibration. Our guarantees will inherit directly from the guarantees of Unbiased-Prediction.

---

**Algorithm 1:** UNBIASED-PREDICTION

---

**for** $t = 1$ **to** $T$ **do**

  Observe $x_t$;

  Define the distribution $q_t \in \Delta[2d|\mathcal{E}|]$ such that for $E \in \mathcal{E}, i \in [d], \sigma \in \{\pm 1\}$,

  $$q_t^{E,i,\sigma} \propto \exp\left(\frac{\eta}{2}\sum_{s=1}^{t-1} \sigma \cdot \mathbb{E}_{p_s \sim \psi_s}[E(x_s, p_s)(p_s^i - y_s^i)]\right);$$

  Output the solution to the minmax problem:

  $$\psi_t \leftarrow \underset{\psi_t' \in \Delta \mathcal{Y}}{\arg\min} \max_{y \in \mathcal{Y}} \mathbb{E}_{p_t \sim \psi_t'}\left[\sum_{E,i,\sigma} q_t^{E,i,\sigma} \cdot \sigma \cdot E(x_t, p_t) \cdot (p_t^i - y_t^i)\right];$$

**end**

---

Our guarantees will hold with high probability $1 - \delta$. To simplify the notation in our bounds, we will define $\zeta = dJ|\mathcal{A}||\mathcal{N}||\mathcal{S}|T/\delta$. Our bounds will depend only logarithmically on it (i.e., as $\ln(\zeta)$).

**Theorem 1.** *Let $\mathcal{S}$ be a collection of subsequences. Let $\mathcal{N}$ be a set of agents, where each agent is equipped with a utility function $u : \mathcal{A} \times \mathcal{Y} \to [0,1]$ and $J$ constraint functions $\{c_j : \mathcal{A} \times \mathcal{Y} \to [-1,1]\}_{j \in [J]}$. There is an instantiation of UNBIASED-PREDICTION (Noarov et al., 2023) —which we call DECISION-INFEASIBILITY-CALIBRATION— producing predictions $p_1, ..., p_T \in \mathcal{Y}$ satisfying that for any sequence of outcomes $y_1, ..., y_T \in \mathcal{Y}$, with probability at least $1 - \delta$, for any $(u, \mathbf{c}) \in \mathcal{N}, j \in [J]$, $a \in \mathcal{A}$, and $S \in \mathcal{S}$:*

$$\left\|\sum_{t=1}^T \mathbb{1}\left[t \in S, \mathrm{CBR}^{u,\mathbf{c}}(p_t) = a\right](p_t - y_t)\right\|_\infty \le O\left(\ln(\zeta) \cdot |S|^{1/4} + \sqrt{\ln(\zeta) \cdot T^{u,\mathbf{c},S}(a)}\right),$$

$$\left\|\sum_{t=1}^T \mathbb{1}\left[t \in S, a \in \widehat{\mathcal{A}}_t^{c_j,\inf}\right](p_t - y_t)\right\|_\infty \le O\left(\ln(\zeta) \cdot |S|^{1/4} + \sqrt{\ln(\zeta) \cdot T^{c_j,S,\inf}(a)}\right).$$

### 3.3 THEORETICAL GUARANTEES

We begin our analysis with a preliminary lemma. Recall that if the set of predicted feasible actions is empty, the agent plays an arbitrary action. The following lemma uses the infeasibility calibration guarantee to show that for any benchmark action $a \in \mathcal{A}_S^{\mathbf{c},\lambda}$, the number of rounds where it is incorrectly predicted to violate a specific constraint $c_j$ is small. As a result, the number of rounds on which no action is predicted to be feasible is small.

First, we establish some notation for simplicity. Let $g_\beta(x) = x/\beta(x)$. For the specific form of $\beta$ provided by the guarantee in Theorem 1, $g_\beta$ is monotone for $x > 0$. We define its inverse function as $f_\beta = g_\beta^{-1}$.

**Lemma 1.** *Suppose the benchmark class $\mathcal{A}_S^{\mathbf{c},\lambda}$ is non-empty. If the sequence of predictions $p_1, \ldots, p_T$ is $(\mathcal{N}, \mathcal{S}, \beta)$-infeasibility calibrated, then for any agent $(u, \mathbf{c}) \in \mathcal{N}$, subsequence $S \in \mathcal{S}$, benchmark action $a \in \mathcal{A}_S^{\mathbf{c},\lambda}$, and constraint $j \in [J]$, the number of rounds $T^{c_j,S,\inf}(a)$ within $S$ on which $a$ is predicted to violate the $j$-th constraint is bounded by:*

$$T^{c_j,S,\inf}(a) \le f_\beta(L_{\mathcal{C}}/\lambda)$$

*Consequently, the number of rounds within $S$ on which no actions are predicted to be feasible is bounded by:*

$$\left|\left\{t \in S : \widehat{\mathcal{A}}_t^{\mathbf{c},\mathrm{fea}} = \emptyset\right\}\right| \le Jf_\beta(L_{\mathcal{C}}/\lambda).$$

*In particular, plugging in the guarantee from Theorem 1 yields the following concrete form of $f_\beta(L_{\mathcal{C}}/\lambda)$, which holds with probability at least $1 - \delta$:*

$$f_\beta(L_{\mathcal{C}}/\lambda) = O\left(\left(\frac{L_{\mathcal{C}}|S|^{1/4}}{\lambda} + \frac{L_{\mathcal{C}}^2}{\lambda^2}\right)\ln(\zeta)\right).$$

### 3.3.1 Bounding the Cumulative Constraint Violation

We now show that the conditional unbiasedness of our predictions guarantees all agents can satisfy the long-term constraints over any subsequences in $\mathcal{S}$. On a high level, this is because (by definition), downstream agents only play actions that we *predict* to be feasible. On the other hand, our predictions are guaranteed to be unbiased on the subsequence of days on which the downstream agents play each action, and so by linearity of the constraint functions, their cumulative constraint violation cannot be much larger than their predicted cumulative constraint violation (which is non-positive).

**Theorem 2.** *Let $\mathcal{S}$ be a collection of subsequences. Let $\mathcal{N}$ be a set of agents, where each agent is equipped with a utility function $u : \mathcal{A} \times \mathcal{Y} \to [0,1]$ and $J$ constraint functions $\{c_j : \mathcal{A} \times \mathcal{Y} \to [-1,1]\}_{j \in [J]}$. Suppose each agent plays constrained best responses to $p_t$ to choose action $a_t$. Suppose the benchmark class $\mathcal{A}_S^{\mathbf{c},\lambda}$ is non-empty. If the sequence of predictions $p_1, \ldots, p_T$ is $(\mathcal{N}, \mathcal{S}, \alpha)$-decision calibrated and $(\mathcal{N}, \mathcal{S}, \beta)$-infeasibility calibrated, then the cumulative constraint violation of any agent over any subsequence $S \in \mathcal{S}$ is bounded by:*

$$\mathrm{CCV}(S) \leq L_{\mathcal{C}}|\mathcal{A}|\alpha(|S|/|\mathcal{A}|) + Jf_{\beta}(L_{\mathcal{C}}/\lambda).$$

*In particular, plugging in the guarantee from Theorem 1 yields the following bound, which holds with probability at least $1 - \delta$:*

$$\mathrm{CCV}(S) \leq O\left(\left(L_{\mathcal{C}}|\mathcal{A}||S|^{1/4} + L_{\mathcal{C}}\sqrt{|\mathcal{A}||S|} + \frac{JL_{\mathcal{C}}|S|^{1/4}}{\lambda} + \frac{JL_{\mathcal{C}}^2}{\lambda^2}\right)\ln(\zeta)\right).$$

We note that this guarantee holds simultaneously for all choice of $\lambda$ (as long as the corresponding benchmark class $\mathcal{A}_S^{\mathbf{c},\lambda}$ is non-empty). By setting $\lambda = |S|^{-1/4}$ for each subsequence $S \in \mathcal{S}$, we arrive at the following concrete bound for the cumulative constraint violation.

**Corollary 1.** *Let $\mathcal{S}$ be a collection of subsequences. Let $\mathcal{N}$ be a set of agents, where each agent is equipped with a utility function $u : \mathcal{A} \times \mathcal{Y} \to [0,1]$ and $J$ constraint functions $\{c_j : \mathcal{A} \times \mathcal{Y} \to [-1,1]\}_{j \in [J]}$. Suppose each agent plays constrained best responses to $p_t$ to compete with actions from the benchmark class $\mathcal{A}_S^{\mathbf{c},|S|^{-1/4}}$ over each subsequence $S \in \mathcal{S}$. The sequence of predictions $p_1, \ldots, p_T$ produced by DECISION-INFEASIBILITY-CALIBRATION guarantees that with probability at least $1 - \delta$, the cumulative constraint violation of any agent over any subsequence $S \in \mathcal{S}$ is bounded by:*

$$\mathrm{CCV}(S) \leq O\left(\left(L_{\mathcal{C}}|\mathcal{A}||S|^{1/4} + L_{\mathcal{C}}\sqrt{|\mathcal{A}||S|} + J(L_{\mathcal{C}} + L_{\mathcal{C}}^2)\sqrt{|S|}\right)\ln(\zeta)\right).$$

**Remark 4.** *Our analysis can be extended to the standard benchmark with zero margin ($\lambda = 0$). This requires relaxing the agent's decision rule to allow for a small tolerance in predicted feasibility (i.e., choosing from actions where $c_j(a, p_t) \leq \eta$). A similar analysis reveals a trade-off, yielding a bound of roughly $\tilde{O}(1/\eta^2 + \eta|S|)$. Optimizing $\eta$ results in cumulative constraint violation and regret bounds of $\tilde{O}(T^{2/3})$. We defer the full results to Appendix C.*

### 3.4 Bounding the Regret

Next we show that decision calibration and infeasibility calibration imply no constrained swap regret, and hence no constrained external regret. At a high level, decision calibrated predictions allow agents to accurately assess (on average) the utilities of their chosen actions and the counterfactual actions produced by any action modification rule. Because agents choose optimally based on these accurate utility estimates, their decisions are competitive against any alternative action as long as the alternative action is also predicted to be feasible. Infeasibility calibrated predictions guarantee that the alternative action must be predicted to be feasible at almost every round.

**Theorem 3.** *Let $\mathcal{S}$ be a collection of subsequences. Let $\lambda : \mathbb{N} \to (0, \infty)$ be a margin function. Let $\mathcal{N}$ be a set of agents, where each agent is equipped with a utility function $u : \mathcal{A} \times \mathcal{Y} \to [0,1]$ and $J$ constraint functions $\{c_j : \mathcal{A} \times \mathcal{Y} \to [-1,1]\}_{j \in [J]}$. Suppose each agent plays constrained best responses to $p_t$ to compete with actions from the benchmark class $\mathcal{A}_S^{\mathbf{c},\lambda}$ over each subsequence $S \in \mathcal{S}$. If the sequence of predictions $p_1, \ldots, p_T$ is $(\mathcal{N}, \mathcal{S}, \alpha)$-decision calibrated and $(\mathcal{N}, \mathcal{S}, \beta)$-infeasibility calibrated, then the constrained swap regret of any agent over any subsequence $S \in \mathcal{S}$ is bounded by:*

$$\mathrm{Reg}_{\mathrm{swap}}(u, \mathbf{c}, \lambda, S) \leq 2L_{\mathcal{U}}|\mathcal{A}|\alpha(|S|/|\mathcal{A}|) + J|\mathcal{A}|f_{\beta}(L_{\mathcal{C}}/\lambda).$$

*In particular, plugging in the guarantee from Theorem 1 yields the following bound, which holds with probability at least $1 - \delta$:*

$$\text{Reg}_{\text{swap}}(u, \mathbf{c}, \lambda, S) \leq O\left(\left(L_{\mathcal{U}}|\mathcal{A}||S|^{1/4} + L_{\mathcal{U}}\sqrt{|\mathcal{A}||S|} + \frac{JL_{\mathcal{C}}|\mathcal{A}||S|^{1/4}}{\lambda} + \frac{JL_{\mathcal{C}}^2|\mathcal{A}|}{\lambda^2}\right)\ln(\zeta)\right).$$

By setting $\lambda = |S|^{-1/4}$ for each subsequence $S \in \mathcal{S}$, we arrive at the following concrete bound for the constrained swap regret.

**Corollary 2.** *Let $\mathcal{S}$ be a collection of subsequences. Let $\mathcal{N}$ be a set of agents, where each agent is equipped with a utility function $u : \mathcal{A} \times \mathcal{Y} \to [0, 1]$ and $J$ constraint functions $\{c_j : \mathcal{A} \times \mathcal{Y} \to [-1, 1]\}_{j \in [J]}$. Suppose each agent plays constrained best responses to $p_t$ to choose action $a_t$. The sequence of predictions $p_1, \ldots, p_T$ produced by DECISION-INFEASIBILITY-CALIBRATION guarantees that with probability at least $1 - \delta$, the constrained swap regret against the benchmark class $\mathcal{A}_S^{\mathbf{c}, |S|^{-1/4}}$ of any agent over any subsequence $S \in \mathcal{S}$ is bounded by:*

$$\text{Reg}_{\text{swap}}(u, \mathbf{c}, |S|^{-1/4}, S) \leq O\left(\left(L_{\mathcal{U}}|\mathcal{A}||S|^{1/4} + L_{\mathcal{U}}\sqrt{|\mathcal{A}||S|} + J(L_{\mathcal{C}} + L_{\mathcal{C}}^2)|\mathcal{A}|\sqrt{|S|}\right)\ln(\zeta)\right).$$

**Remark 5.** *One could alternatively set a fixed margin of $\lambda = T^{-1/4}$. This choice creates a larger and more competitive benchmark class. The resulting regret bound would then be $\tilde{O}(\sqrt{T})$.*

A powerful implication is that we achieve low constrained swap adaptive regret by instantiating our framework with the collection of all contiguous intervals, $\mathcal{S} = \{[t_1, t_2] : 1 \leq t_1 \leq t_2 \leq T\}$.

**Corollary 3.** *Let $\mathcal{N}$ be a set of agents, where each agent is equipped with a utility function $u : \mathcal{A} \times \mathcal{Y} \to [0, 1]$ and $J$ constraint functions $\{c_j : \mathcal{A} \times \mathcal{Y} \to [-1, 1]\}_{j \in [J]}$. Suppose each agent plays constrained best responses to $p_t$ to choose action $a_t$. Let $\lambda(|S|) = |S|^{-1/4}$ be the margin function. The sequence of predictions $p_1, \ldots, p_T$ produced by DECISION-INFEASIBILITY-CALIBRATION guarantees that with probability at least $1 - \delta$, the constrained swap adaptive regret of any agent is bounded by:*

$$\text{Reg}_{\text{swap-adapt}}(u, \mathbf{c}, \lambda) \leq O\left(\left(L_{\mathcal{U}}|\mathcal{A}|T^{1/4} + L_{\mathcal{U}}\sqrt{|\mathcal{A}|T} + J(L_{\mathcal{C}} + L_{\mathcal{C}}^2)|\mathcal{A}|\sqrt{T}\right)\ln(\zeta)\right).$$

Corollary 3 provides a guarantee over all contiguous intervals, and we use it as a building block to establish dynamic regret bounds. A dynamic benchmark with $\Delta$ changes partitions the entire time horizon into $\Delta + 1$ intervals. By summing our per-subsequence regret bound over this specific partition, we obtain the following dynamic regret guarantee.

**Corollary 4.** *Let $\mathcal{N}$ be a set of agents, where each agent is equipped with a utility function $u : \mathcal{A} \times \mathcal{Y} \to [0, 1]$ and $J$ constraint functions $\{c_j : \mathcal{A} \times \mathcal{Y} \to [-1, 1]\}_{j \in [J]}$. Suppose each agent plays constrained best responses to $p_t$ to choose action $a_t$. The sequence of predictions $p_1, \ldots, p_T$ produced by DECISION-INFEASIBILITY-CALIBRATION guarantees that with probability at least $1 - \delta$, the constrained swap dynamic regret of any agent against any piecewise feasible sequence of action modification rule $\vec{\phi} \in (\mathcal{A}^{\mathcal{A}})^T$ is bounded by:*

$$\text{Reg}_{\text{swap-dyn}}(u, \vec{\phi}) \leq O\left(\left(L_{\mathcal{U}}|\mathcal{A}|T^{\frac{1}{4}}\Delta(\vec{\phi})^{\frac{3}{4}} + L_{\mathcal{U}}\sqrt{|\mathcal{A}|T\Delta(\vec{\phi})} + J(L_{\mathcal{C}} + L_{\mathcal{C}}^2)|\mathcal{A}|\sqrt{T\Delta(\vec{\phi})}\right)\ln(\zeta)\right).$$

*A benchmark sequence $\vec{\phi}$ is piecewise feasible if on each interval of constancy $I_k$, the corresponding rule $\psi_k$ maps to the set of actions that are feasible over that entire interval, i.e., $\psi_k : \mathcal{A} \to \mathcal{A}_{I_k}^{\mathbf{c}, |I_k|^{-1/4}}$.*

This bound explicitly characterizes the dependence of the regret on the time horizon $T$ and the complexity of the comparator sequence, denoted by the number of switches $\Delta(\vec{\phi})$. In the stationary regime where $\Delta(\vec{\phi}) = 0$, we recover the standard $\tilde{O}(\sqrt{T})$ rate for static regret; in the dynamic regime where $\Delta(\vec{\phi}) = o(T)$, the bound ensures sublinear regret. Crucially, this guarantee is parameter-free: the algorithm does not need to know the value of $\Delta(\vec{\phi})$ in advance. This is achieved by instantiating our framework with the collection of all contiguous intervals.

Since external regret is a special case of swap regret, our bounds apply directly to the external versions of these guarantees as well.

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

## A    ADDITIONAL RELATED WORK

The study of online learning with long-term constraints was initiated by Mannor et al. (2009), who established a foundational impossibility result. They demonstrated that learners cannot achieve sublinear external regret when benchmarked against the set of actions that satisfy the constraints marginally, i.e., on average over the entire time horizon. In response to this limitation, subsequent work pivoted to a more stringent benchmark: actions that fulfill the constraints on every single round (e.g., Sun et al. (2017); Guo et al. (2022); Anderson et al. (2022); Yi et al. (2023); Sinha & Vaze (2024); Lekeufack & Jordan (2024)). These papers generally study the problem within the online convex optimization framework, rather than for a finite number of actions. Within this body of work, Sinha & Vaze (2024) achieved the fastest known rates without additional structural assumptions, with both regret and cumulative constraint violation scaling as $\tilde{O}(\sqrt{T})$. For the experts setting,

Lekeufack & Jordan (2024) proposed an algorithm where both metrics scale as $\tilde{O}(\sqrt{T \ln(d)})$, with $d$ representing the number of experts and regret measured against the set of expert probability distributions that satisfy the constraints in expectation.

This literature has also studied dynamic regret benchmarks (Chen et al. (2017; 2018); Chen & Giannakis (2018); Cao & Liu (2019); Vaze (2022); Liu et al. (2022b); Lekeufack & Jordan (2024)), where regret is measured against a slowly changing sequence of actions, one for every round in the interaction. We introduce the stronger notion of dynamic swap regret.

Some papers in this literature (e.g. Neely & Yu (2017); Chen et al. (2017); Cao & Liu (2019); Yu & Neely (2020); Castiglioni et al. (2022)) assume Slater's condition holds — the existence of a benchmark action that satisfies all of the constraints with constant margin (or a sequence of strongly feasible actions, in the dynamic benchmark case), which is an assumption we sometimes make in this work as well. Note that we also provide bounds that hold without this assumption.

Omniprediction was introduced by Gopalan et al. (2022) within the batch (distributional) learning setting. In its standard formulation, omniprediction considers a binary label space $\mathcal{Y} = \{0, 1\}$, a continuous action space of real-valued predictions $\mathcal{A} = [0, 1]$, and decision-makers who optimize a loss function $\ell : [0, 1] \times \{0, 1\} \to \mathbb{R}$, such as squared or absolute error. A key initial finding was that multicalibration (c.f. Hébert-Johnson et al. (2018)) is a sufficient condition for achieving omniprediction. Subsequent work (Gopalan et al., 2023a;b) studied both weaker and stronger notions of calibration and the corresponding omniprediction guarantees they provide.

Omniprediction was later extended to the online adversarial setting by Garg et al. (2024), who provided oracle-efficient algorithms, with subsequent work establishing optimal regret bounds (Oko-roafor et al., 2025; Dwork et al., 2025). While the bulk of this literature focuses on binary outcomes, a notable exception is Gopalan et al. (2024), which investigates vector-valued outcomes for decision-makers with convex loss functions. The problem of omniprediction with constraints has been explored in (Globus-Harris et al., 2023; Hu et al., 2023). Consistent with the broader literature, these works operate in the batch setting with binary outcomes and continuous actions, focusing on constraints motivated by group fairness in machine learning classification tasks.

A parallel line of work has emerged on making sequential predictions for downstream agents Klein-berg et al. (2023); Noarov et al. (2023); Roth & Shi (2024); Hu & Wu (2024). This literature considers adversarially chosen outcomes from a space $\mathcal{Y}$ and downstream agents who possess arbitrary discrete action spaces and seek to optimize their own losses. Several of these papers move beyond the binary setting Noarov et al. (2023); Roth & Shi (2024), addressing scenarios—similar to our own—where loss functions are linear in the high-dimensional state $y \in [0, 1]^d$ to be predicted. The primary objective is to provide worst-case, diminishing regret guarantees for all such agents. Motivated by competitive environments, a subset of this work Noarov et al. (2023); Roth & Shi (2024); Hu & Wu (2024) provides guarantees for diminishing swap regret at near-optimal rates.

Recognizing that traditional calibration is unobtainable at $O(\sqrt{T})$ rates in the online adversarial setting (Qiao & Valiant, 2021; Dagan et al., 2024), this literature has employed alternative techniques. Methods such as "U-calibration" (Kleinberg et al., 2023) and extensions of decision calibration Zhao et al. (2021); Noarov et al. (2023); Roth & Shi (2024) circumvent these known lower bounds while still being powerful enough to ensure downstream agents incur no (swap) regret. A unifying perspective on these two research areas is offered by Lu et al. (2025). Our own model and techniques are primarily derived from this "prediction for downstream regret" literature, adopting its features of arbitrary action spaces, high-dimensional outcomes, and swap regret guarantees in an online adversarial context.

Finally, our work is a direct follow up to Bechavod et al. (2025) who introduced the problem of "omniprediction with long term constraints". Like Bechavod et al. (2025), we give an algorithm that can guarantee an arbitrary collection of decision makers diminishing regret bounds on an arbitrarily specified collection of subsequences. Our main technical contribution is an exponentially improved dependence (in both the regret and constraint violation terms) on the number of such subsequences — the algorithm of Bechavod et al. (2025) depends *linearly* on this, whereas our bounds depend only *logarithmically* on this parameter. This is crucial in order to give dynamic regret bounds, which fall out of guaranteeing diminishing (swap) regret on all contiguous subsequences, of which there are $\Omega(T^2)$. This would yield trivial bounds using the algorithm of Bechavod et al. (2025), whereas it

costs only an additional logarithmic term in our regret bounds relative to what we could guarantee for a static benchmark.

## B  OMITTED PROOFS

### B.1  PROOF OF LEMMA 1

On any round $t$ where action $a$ is predicted to violate the $j$-th constraint, we have:

$$c_j(a, p_t) > 0$$

Since $a$ is in the benchmark class $\mathcal{A}_S^{\mathbf{c}, \lambda}$, we have:

$$c_j(a, y_t) \leq -\lambda$$

Combining these two facts gives $c_j(a, p_t) - c_j(a, y_t) > \lambda$. Summing this difference over all rounds in $S$ where $a$ is predicted to violate the $j$-th constraint, we get:

$$\sum_{t=1}^{T} \mathbb{1}\left[t \in S, a \in \widehat{\mathcal{A}}_t^{c_j, \mathrm{inf}}\right] (c_j(a, p_t) - c_j(a, y_t)) > \lambda T^{c_j, S, \mathrm{inf}}(a)$$

The left-hand side can be bounded using the properties of our predictions. By linearity and $L_{\mathcal{C}}$-Lipschitzness of the constraint function, and by $(\mathcal{N}, \mathcal{S}, \beta)$-infeasibility calibration, we have that:

$$\sum_{t=1}^{T} \mathbb{1}\left[t \in S, a \in \widehat{\mathcal{A}}_t^{c_j, \mathrm{inf}}\right] (c_j(a, p_t) - c_j(a, y_t))$$

$$= c_j\left(a, \sum_{t=1}^{T} p_t \mathbb{1}\left[t \in S, a \in \widehat{\mathcal{A}}_t^{c_j, \mathrm{inf}}\right]\right) - c_j\left(a, \sum_{t=1}^{T} y_t \mathbb{1}\left[t \in S, a \in \widehat{\mathcal{A}}_t^{c_j, \mathrm{inf}}\right]\right)$$

$$\leq L_{\mathcal{C}}\left\|\sum_{t=1}^{T} p_t \mathbb{1}\left[t \in S, a \in \widehat{\mathcal{A}}_t^{c_j, \mathrm{inf}}\right] - \sum_{t=1}^{T} y_t \mathbb{1}\left[t \in S, a \in \widehat{\mathcal{A}}_t^{c_j, \mathrm{inf}}\right]\right\|_{\infty}$$

$$\leq L_{\mathcal{C}} \beta(T^{c_j, S, \mathrm{inf}}(a))$$

Combining these inequalities gives the first part of the lemma:

$$\lambda T^{c_j, S, \mathrm{inf}}(a) < L_{\mathcal{C}} \beta(T^{c_j, S, \mathrm{inf}}(a))$$

For the concrete bound, we substitute the explicit form for $\beta$ from Theorem 1:

$$\lambda f_\beta(L_{\mathcal{C}}/\lambda) = L_{\mathcal{C}} O\left(\ln \frac{dJ|\mathcal{A}||\mathcal{N}||\mathcal{S}|T}{\delta} \cdot |S|^{1/4} + \sqrt{\ln \frac{dJ|\mathcal{A}||\mathcal{N}||\mathcal{S}|T}{\delta} \cdot f_\beta(L_{\mathcal{C}}/\lambda)}\right)$$

Solving for $f_\beta(L_{\mathcal{C}}/\lambda)$ yields the stated form:

$$f_\beta(L_{\mathcal{C}}/\lambda) = O\left(\frac{L_{\mathcal{C}}|S|^{1/4}\ln(dJ|\mathcal{A}||\mathcal{N}||\mathcal{S}|T/\delta)}{\lambda} + \frac{L_{\mathcal{C}}^2 \ln(dJ|\mathcal{A}||\mathcal{N}||\mathcal{S}|T/\delta)}{\lambda^2}\right)$$

Fix any arbitrary benchmark action $a \in \mathcal{A}_S^{\mathbf{c}, \lambda}$. If on round $t \in S$, no action is predicted to be feasible, then it must be that $a$ is predicted to be infeasible. Hence, there must exist at least one constraint that $a$ is predicted to violate. As a result, we have:

$$\left|\left\{t \in S : \widehat{\mathcal{A}}_t^{\mathbf{c}, \mathrm{fea}} = \emptyset\right\}\right| \leq |\{t \in S : \exists j \in [J], c_j(a, p_t) > 0\}|$$

$$\leq \sum_{j=1}^{J} T^{c_j, S, \mathrm{inf}}(a)$$

$$\leq J f_\beta(L_{\mathcal{C}}/\lambda)$$

## B.2 Proof of Theorem 2

Fix any $j \in [J]$. We will bound the cumulative violation against the $j$-th constraint, i.e., $\sum_{t \in S} c_j(a_t, y_t)$. The final result follows by taking the maximum over all $j \in [J]$.

We use the triangle inequality to decompose the sum:

$$\sum_{t \in S} c_j(a_t, y_t) \leq \left| \sum_{t \in S} c_j(a_t, y_t) - \sum_{t \in S} c_j(a_t, p_t) \right| + \sum_{t \in S} c_j(a_t, p_t)$$

We bound each of the two terms on the right-hand side separately.

For the first term, we partition the sum based on the action played and use the linearity of the constraint functions to derive that:

$$\left| \sum_{t \in S} c_j(a_t, y_t) - \sum_{t \in S} c_j(a_t, p_t) \right| = \left| \sum_{a \in \mathcal{A}} \sum_{t \in S} c_j(a, y_t) \mathbb{1}[a_t = a] - \sum_{a \in \mathcal{A}} \sum_{t \in S} c_j(a, p_t) \mathbb{1}[a_t = a] \right|$$

$$= \left| \sum_{a \in \mathcal{A}} c_j \left( a, \sum_{t \in S} y_t \mathbb{1}[a_t = a] \right) - \sum_{a \in \mathcal{A}} c_j \left( a, \sum_{t \in S} p_t \mathbb{1}[a_t = a] \right) \right|$$

$$\leq \sum_{a \in \mathcal{A}} \left| c_j \left( a, \sum_{t \in S} y_t \mathbb{1}[a_t = a] \right) - c_j \left( a, \sum_{t \in S} p_t \mathbb{1}[a_t = a] \right) \right|$$

$$\leq \sum_{a \in \mathcal{A}} L_{\mathcal{C}} \left| \sum_{t \in S} y_t \mathbb{1}[a_t = a] - \sum_{t \in S} p_t \mathbb{1}[a_t = a] \right|$$

$$\leq \sum_{a \in \mathcal{A}} L_{\mathcal{C}} \alpha(T^{u,\mathbf{c},S}(a))$$

where the first inequality follows from the triangle inequality, the second inequality follows from $L_{\mathcal{C}}$-Lipschitzness of $c_j$, the third inequality follows from $(\mathcal{N}, \mathcal{S}, \alpha)$-decision calibration. By concavity of $\alpha$ and the fact that $\sum_{a \in \mathcal{A}} T^{u,\mathbf{c},S}(a) = \sum_{a \in \mathcal{A}} \sum_{t=1}^{T} \mathbb{1}\left[ t \in S, \mathrm{CBR}^{u,\mathbf{c}}(p_t) = a \right] = |S|$, this expression is at most:

$$L_{\mathcal{C}} |\mathcal{A}| \alpha(|S|/|\mathcal{A}|).$$

For the second term, $\sum_{t \in S} c_j(a_t, p_t)$, we decompose the sum based on whether the predicted feasible set $\widehat{\mathcal{A}}_t^{\mathbf{c},\mathrm{fea}}$ is empty on round $t$. By the agent's decision rule, on any round where $\widehat{\mathcal{A}}_t^{\mathbf{c},\mathrm{fea}} \neq \emptyset$, the chosen action $a_t$ satisfies $c_j(a_t, p_t) \leq 0$. On rounds where the set is empty, the violation is at most 1. The sum is therefore bounded by the number of "empty set" rounds:

$$\sum_{t \in S} c_j(a_t, p_t) = \sum_{t \in S : \widehat{\mathcal{A}}_t^{\mathbf{c},\mathrm{fea}} \neq \emptyset} c_j(a_t, p_t) + \sum_{t \in S : \widehat{\mathcal{A}}_t^{\mathbf{c},\mathrm{fea}} = \emptyset} c_j(a_t, p_t)$$

$$\leq \sum_{t \in S : \widehat{\mathcal{A}}_t^{\mathbf{c},\mathrm{fea}} \neq \emptyset} 0 + \sum_{t \in S : \widehat{\mathcal{A}}_t^{\mathbf{c},\mathrm{fea}} = \emptyset} 1$$

$$= \left| \left\{ t \in S : \widehat{\mathcal{A}}_t^{\mathbf{c},\mathrm{fea}} = \emptyset \right\} \right|$$

By Lemma 1, this expression is at most:

$$J f_\beta(L_{\mathcal{C}}/\lambda)$$

Combining the two bounds gives the result stated in the theorem.

## B.3 Proof of Theorem 3

To prove the theorem, first note that for any subsequence $S \in \mathcal{S}$ and action modification rule $\phi : \mathcal{A} \to \mathcal{A}_S^{\mathbf{c}}$, we can decompose the regret against $\phi$ into three parts as:

$$\sum_{t \in S} (u(\phi(a_t), y_t) - u(a_t, y_t))$$

$$= \sum_{t \in S} (u(\phi(a_t), y_t) - u(\phi(a_t), p_t)) + \sum_{t \in S} (u(\phi(a_t), p_t) - u(a_t, p_t)) + \sum_{t \in S} (u(a_t, p_t) - u(a_t, y_t))$$

We first bound the first and third part, i.e., the difference in utility under our predictions $p_t$ and the outcomes $y_t$ for both the chosen actions and the swapped-in actions. We show this in the next two lemmas using decision calibration.

**Lemma 2.** *If the sequence of predictions $p_1, \ldots, p_T$ is $(\mathcal{N}, \mathcal{S}, \alpha)$-decision calibrated, then for any $(u, \mathbf{c}) \in \mathcal{N}$ and $S \in \mathcal{S}$:*

$$\left| \sum_{t \in S} (u(a_t, p_t) - u(a_t, y_t)) \right| \leq L_{\mathcal{U}} |\mathcal{A}| \alpha(|S|/|\mathcal{A}|).$$

*Proof.* Using the linearity of $u$, we can write:

$$\left| \sum_{t \in S} (u(a_t, p_t) - u(a_t, y_t)) \right| = \left| \sum_{a \in \mathcal{A}} \sum_{t=1}^{T} \mathbb{1}[t \in S, a_t = a](u(a, p_t) - u(a, y_t)) \right|$$

$$= \left| \sum_{a \in \mathcal{A}} \left( u\left(a, \sum_{t=1}^{T} \mathbb{1}[t \in S, a_t = a]p_t\right) - u\left(a, \sum_{t=1}^{T} \mathbb{1}[t \in S, a_t = a]y_t\right) \right) \right|$$

$$\leq \sum_{a \in \mathcal{A}} \left| u\left(a, \sum_{t=1}^{T} \mathbb{1}[t \in S, a_t = a]p_t\right) - u\left(a, \sum_{t=1}^{T} \mathbb{1}[t \in S, a_t = a]y_t\right) \right|$$

$$\leq \sum_{a \in \mathcal{A}} L_{\mathcal{U}} \left\| \sum_{t=1}^{T} \mathbb{1}[t \in S, a_t = a](p_t - y_t) \right\|_{\infty}$$

$$\leq L_{\mathcal{U}} \sum_{a \in \mathcal{A}} \alpha(T^{u, \mathbf{c}, S}(a)).$$

where the first inequality follows from the triangle inequality, the second inequality follows from $L_{\mathcal{U}}$-Lipschitzness of $u$, and the third inequality follows from $(\mathcal{N}, \mathcal{S}, \alpha)$-decision calibration. By concavity of $\alpha$ and the fact that $\sum_{a \in \mathcal{A}} T^{u, \mathbf{c}, S}(a) = \sum_{a \in \mathcal{A}} \sum_{t=1}^{T} \mathbb{1}[t \in S, a_t = a] = |S|$, this expression is at most:

$$L_{\mathcal{U}} |\mathcal{A}| \alpha(|S|/|\mathcal{A}|).$$

$\square$

**Lemma 3.** *If the sequence of predictions $p_1, \ldots, p_T$ is $(\mathcal{N}, \mathcal{S}, \alpha)$-decision calibrated, then for any $(u, \mathbf{c}) \in \mathcal{N}$ and $S \in \mathcal{S}$:*

$$\left| \sum_{t \in S} (u(\phi(a_t), p_t) - u(\phi(a_t), y_t)) \right| \leq L_{\mathcal{U}} |\mathcal{A}| \alpha(|S|/|\mathcal{A}|).$$

*Proof.* Using the linearity of $u$, we can write:

$$\left| \sum_{t \in S} (u(\phi(a_t), p_t) - u(\phi(a_t), y_t)) \right| = \left| \sum_{a \in \mathcal{A}} \sum_{t=1}^{T} \mathbb{1}[t \in S, a_t = a](u(\phi(a), p_t) - u(\phi(a), y_t)) \right|$$

$$= \left| \sum_{a \in \mathcal{A}} \left( u\left(\phi(a), \sum_{t=1}^{T} \mathbb{1}[t \in S, a_t = a]p_t\right) - u\left(\phi(a), \sum_{t=1}^{T} \mathbb{1}[t \in S, a_t = a]y_t\right) \right) \right|$$

$$\leq \sum_{a \in \mathcal{A}} \left| u\left(\phi(a), \sum_{t=1}^{T} \mathbb{1}[t \in S, a_t = a]p_t\right) - u\left(\phi(a), \sum_{t=1}^{T} \mathbb{1}[t \in S, a_t = a]y_t\right) \right|$$

$$\leq \sum_{a \in \mathcal{A}} L_{\mathcal{U}} \left\| \sum_{t=1}^{T} \mathbb{1}[t \in S, a_t = a](p_t - y_t) \right\|_{\infty}$$

$$\leq L_{\mathcal{U}} \sum_{a \in \mathcal{A}} \alpha(T^{u,\mathbf{c},S}(a))$$

$$\leq L_{\mathcal{U}} |\mathcal{A}| \alpha(|S|/|\mathcal{A}|)$$

where the first inequality follows from the triangle inequality, the second inequality follows from $L_{\mathcal{U}}$-Lipschitzness of $u$, the third inequality follows from $(\mathcal{N}, \mathcal{S}, \alpha)$-decision calibration, and the fourth inequality follows from concavity of $\alpha$. $\qquad\square$

Regarding the second part in our decomposition of the regret against $\phi$, we further decompose it into two components based on whether the swapped-in action is predicted to be feasible:

$$\sum_{t \in S} (u(\phi(a_t), p_t) - u(a_t, p_t)) = \sum_{t \in S: \phi(a_t) \in \widehat{\mathcal{A}}_t^{\mathbf{c}, \text{fea}}} (u(\phi(a_t), p_t) - u(a_t, p_t))$$

$$+ \sum_{t \in S: \phi(a_t) \in \widehat{\mathcal{A}}_t^{\mathbf{c}, \text{inf}}} (u(\phi(a_t), p_t) - u(a_t, p_t))$$

The first component is non-positive. This is because on these rounds, $a_t$ is chosen to maximize predicted utility over the predicted feasible set, which includes $\phi(a_t)$, so $u(\phi(a_t), p_t) \leq u(a_t, p_t)$.

For the second component, the utility difference is at most 1, so the sum is bounded by the number of rounds within $S$ on which the swapped-in action $\phi(a_t)$ is predicted to be infeasible. We can bound this count by the sum of the number of rounds where each single benchmark action $a \in \mathcal{A}_S^{\mathbf{c},\lambda}$ is predicted to be infeasible.

$$\sum_{t \in S: \phi(a_t) \in \widehat{\mathcal{A}}_t^{\mathbf{c}, \text{inf}}} (u(\phi(a_t), p_t) - u(a_t, p_t)) \leq \sum_{t \in S} \mathbb{1}\left[\phi(a_t) \in \widehat{\mathcal{A}}_t^{\mathbf{c}, \text{inf}}\right]$$

$$= \sum_{a \in \mathcal{A}_S^{\mathbf{c},\lambda}} \sum_{t \in S} \mathbb{1}\left[\phi(a_t) = a, a \in \widehat{\mathcal{A}}_t^{\mathbf{c}, \text{inf}}\right]$$

$$\leq \sum_{a \in \mathcal{A}_S^{\mathbf{c},\lambda}} \sum_{t \in S} \mathbb{1}\left[a \in \widehat{\mathcal{A}}_t^{\mathbf{c}, \text{inf}}\right]$$

$$= \sum_{a \in \mathcal{A}_S^{\mathbf{c},\lambda}} \sum_{t \in S} \mathbb{1}\left[\exists j \in [J], c_j(a, p_t) > 0\right]$$

$$\leq \sum_{a \in \mathcal{A}_S^{\mathbf{c},\lambda}} \sum_{t \in S} \sum_{j=1}^{J} \mathbb{1}\left[c_j(a, p_t) > 0\right]$$

$$= \sum_{a \in \mathcal{A}_S^{\mathbf{c},\lambda}} \sum_{j=1}^{J} T^{c_j, S, \text{inf}}(a)$$

By Lemma 1, this expression is at most:

$$J|\mathcal{A}| f_\beta(L_{\mathcal{C}}/\lambda)$$

We can now complete the proof of Theorem 3. For any subsequence $S \in \mathcal{S}$ and action modification rule $\phi : \mathcal{A} \to \mathcal{A}_S^{\mathbf{c}}$, we can decompose the regret against $\phi$ into three parts as:

$$\sum_{t \in S} (u(\phi(a_t), y_t) - u(a_t, y_t))$$

$$= \sum_{t \in S} (u(\phi(a_t), y_t) - u(\phi(a_t), p_t)) + \sum_{t \in S} (u(\phi(a_t), p_t) - u(a_t, p_t)) + \sum_{t \in S} (u(a_t, p_t) - u(a_t, y_t))$$

By Lemmas 2 and 3, the first and third part are both bounded by $L_{\mathcal{U}} |\mathcal{A}| \alpha(|S|/|\mathcal{A}|)$.

As shown in our preceding analysis, the second part is bounded by $J|\mathcal{A}| f_\beta(L_{\mathcal{C}}/\lambda)$.

Combining the bounds for all three parts, we arrive at the final inequality.

### B.4 Proof of Corollary 4

Suppose the sequence $\vec{\phi}$ have change points that partion $[1, T]$ into intervals $I_0, \ldots, I_{\Delta(\vec{\phi})}$. The action modification rule on each interval $I_k$ is the constant rule $\psi_k : \mathcal{A} \to \mathcal{A}_{I_k}^{\mathbf{c}, \lambda}$.

By Theorem 3, for any interval $I_k$, the regret against $\psi_k$ over $I_k$ is bounded by:

$$\sum_{t=\min\{I_k\}}^{\max\{I_k\}} (u(\psi_k(a_t), y_t) - u(a_t, y_t))$$

$$\leq O\left(\left(L_{\mathcal{U}}|\mathcal{A}||I_k|^{1/4} + L_{\mathcal{U}}\sqrt{|\mathcal{A}||I_k|} + \frac{JL_{\mathcal{C}}|\mathcal{A}||I_k|^{1/4}}{\lambda} + \frac{JL_{\mathcal{C}}^2|\mathcal{A}|}{\lambda^2}\right)\ln(dJ|\mathcal{A}||\mathcal{N}||\mathcal{S}|T/\delta)\right)$$

Summing over all the intervals, we have:

$$\sum_{t=1}^{T}(u(\psi_k(a_t), y_t) - u(a_t, y_t))$$

$$= \sum_{k=0}^{\Delta(\vec{\phi})} \sum_{t=\min\{I_k\}}^{\max\{I_k\}} (u(\psi_k(a_t), y_t) - u(a_t, y_t))$$

$$\leq \sum_{k=0}^{\Delta(\vec{\phi})} O\left(\left(L_{\mathcal{U}}|\mathcal{A}||I_k|^{1/4} + L_{\mathcal{U}}\sqrt{|\mathcal{A}||I_k|} + \frac{JL_{\mathcal{C}}|\mathcal{A}||I_k|^{1/4}}{\lambda} + \frac{JL_{\mathcal{C}}^2|\mathcal{A}|}{\lambda^2}\right)\ln(dJ|\mathcal{A}||\mathcal{N}||\mathcal{S}|T/\delta)\right)$$

We note that $\lambda$ can be set independently for each interval $I_k$. By setting $\lambda = |I_k|^{-1/4}$, we arrive at the following bound:

$$\sum_{k=0}^{\Delta(\vec{\phi})} O\left(\left(L_{\mathcal{U}}|\mathcal{A}||I_k|^{1/4} + L_{\mathcal{U}}\sqrt{|\mathcal{A}||I_k|} + J(L_{\mathcal{C}} + L_{\mathcal{C}}^2)|\mathcal{A}|\sqrt{|I_k|}\right)\ln(dJ|\mathcal{A}||\mathcal{N}||\mathcal{S}|T/\delta)\right)$$

By concavity of the functions $f_1(x) = x^{1/4}$ and $f_2(x) = x^{1/2}$ and the fact that $\sum_{k=0}^{\Delta(\vec{\phi})} |I_k| = T$, this expression is at most:

$$O\left(\left(L_{\mathcal{U}}|\mathcal{A}|T^{1/4}\Delta(\vec{\phi})^{3/4} + L_{\mathcal{U}}\sqrt{|\mathcal{A}|T\Delta(\vec{\phi})} + J(L_{\mathcal{C}} + L_{\mathcal{C}}^2)|\mathcal{A}|\sqrt{T\Delta(\vec{\phi})}\right)\ln(dJ|\mathcal{A}||\mathcal{N}||\mathcal{S}|T/\delta)\right)$$

## C  Analysis for the Zero-Margin Benchmark

### C.1  A Decision Rule with Feasibility Tolerance

In this section, we obtain guarantees for the standard benchmark $\mathcal{A}_S^{\mathbf{c}, 0}$ with zero margin $\lambda = 0$. For this purpose, we modify the agent's decision rule. We introduce a relaxed rule that allows for a small, positive tolerance in predicted feasibility $\eta > 0$. An action is now considered feasible by the agent if its predicted constraint violation is less than or equal to this tolerance.

Formally, we say that an action $a$ is predicted to $\eta$-violate the $j$-th constraint if $c_j(a, p_t) > \eta$. An action $a$ is predicted to be $\eta$-infeasible if $a$ is predicted to $\eta$-violate any constraint, and is predicted to be $\eta$-feasible otherwise. This leads to a relaxed version of the constrained best response.

**Definition 8** ($\eta$-Constrained Best Response). *Fix a utility* $u : \mathcal{A} \times \mathcal{Y} \to [0, 1]$, *J constraints* $\{c_j : \mathcal{A} \times \mathcal{Y} \to [-1, 1]\}_{j \in [J]}$, *a prediction* $p \in \mathcal{Y}$, *and a tolerance* $\eta > 0$. *The* $\eta$-*constrained best response to* $p$ *according to* $u$ *and* $\{c_j\}_{j \in [J]}$, *denoted as* $\mathrm{CBR}_\eta^{u, \mathbf{c}}(p)$, *is the solution to the constrained optimization problem:*

$$\begin{aligned} \underset{a \in \mathcal{A}}{\text{maximize}} \quad & u(a, p_t) \\ \text{subject to} \quad & c_j(a, p_t) \leq \eta \text{ for every } j \in [J] \end{aligned}$$

The agent can obtain $\mathrm{CBR}_\eta^{u,\mathbf{c}}(p_t)$ in two steps:

(1) The agent first discards actions that are $\eta$-infeasible according to the prediction.
    For each constraint $j \in [J]$, the set of actions predicted to $\eta$-violate that constraint is denoted as:

$$\widehat{\mathcal{A}}_t^{c_j,\eta-\mathrm{inf}} = \{a \in \mathcal{A} : c_j(a, p_t) > \eta\}.$$

The agent discards actions that are predicted to $\eta$-violate any of the $J$ constraints, i.e.,

$$\widehat{\mathcal{A}}_t^{\mathbf{c},\eta-\mathrm{inf}} = \cup_{j \in [J]} \widehat{\mathcal{A}}_t^{c_j,\eta-\mathrm{inf}} = \{a \in \mathcal{A} : \exists j \in [J], c_j(a, p_t) > \eta\}.$$

The retained actions that are predicted to be $\eta$-feasible are denoted as

$$\widehat{\mathcal{A}}_t^{\mathbf{c},\eta-\mathrm{fea}} = \{a \in \mathcal{A} : \forall j \in [J], c_j(a, p_t) \le \eta\}.$$

(2) The agent then chooses an action from the retained action set $\widehat{\mathcal{A}}_t^{\mathbf{c},\eta-\mathrm{fea}}$ that maximizes the utility function according to the prediction, i.e.,

$$a_t = \underset{a \in \widehat{\mathcal{A}}_t^{\mathbf{c},\eta-\mathrm{fea}}}{\arg\max}\ u(a, p_t).$$

If none of the actions are predicted to be feasible, i.e., $\widehat{\mathcal{A}}_t^{\mathbf{c},\eta-\mathrm{fea}} = \emptyset$, the agent can choose any arbitrary action from $\mathcal{A}$. Our predictions ensure this special case rarely occurs, and hence its influence on the cumulative constraint violation and regret is negligible.

### C.2 CONDITIONALLY UNBIASED PREDICTIONS

As in our main analysis, the guarantees for the relaxed decision rule rely on the predictions being conditionally unbiased. The definitions for decision calibration and infeasibility calibration are analogous to those in the main text, modified to account for the feasibility tolerance $\eta$.

**Definition 9** $((\mathcal{N}, \mathcal{S}, \eta, \alpha)$-Decision Calibration**).** *Let $\mathcal{S}$ be a collection of subsequences. Let $\mathcal{N}$ be a set of agents, where each agent is equipped with a utility function $u : \mathcal{A} \times \mathcal{Y} \to [0, 1]$ and $J$ constraint functions $\{c_j : \mathcal{A} \times \mathcal{Y} \to [-1, 1]\}_{j \in [J]}$. Let $\eta > 0$ be the feasibility tolerance. We say that a sequence of predictions $p_1, \ldots, p_T$ is $(\mathcal{N}, \mathcal{S}, \eta, \alpha)$-decision calibrated with respect to a sequence of outcomes $y_1, \ldots, y_T$ if for every $S \in \mathcal{S}$, $a \in \mathcal{A}$, and $(u, \mathbf{c}) \in \mathcal{N}$:*

$$\left\| \sum_{t=1}^T \mathbb{1}\left[t \in S, \mathrm{CBR}_\eta^{u,\mathbf{c}}(p_t) = a\right] (p_t - y_t) \right\|_\infty \le \alpha(T^{u,\mathbf{c},S,\eta}(a))$$

*where $T^{u,\mathbf{c},S,\eta}(a) = \sum_{t=1}^T \mathbb{1}\left[t \in S, \mathrm{CBR}_\eta^{u,\mathbf{c}}(p_t) = a\right]$.*

**Definition 10** $((\mathcal{N}, \mathcal{S}, \eta, \beta)$-Infeasibility Calibration**).** *Let $\mathcal{S}$ be a collection of subsequences. Let $\mathcal{N}$ be a set of agents, where each agent is equipped with a utility function $u : \mathcal{A} \times \mathcal{Y} \to [0, 1]$ and $J$ constraint functions $\{c_j : \mathcal{A} \times \mathcal{Y} \to [-1, 1]\}_{j \in [J]}$. Let $\eta > 0$ be the feasibility tolerance. We say that a sequence of predictions $p_1, \ldots, p_T$ is $(\mathcal{N}, \mathcal{S}, \eta, \beta)$-infeasibility calibrated with respect to a sequence of outcomes $y_1, \ldots, y_T$ if for every $S \in \mathcal{S}$, $a \in \mathcal{A}$, $(u, \mathbf{c}) \in \mathcal{N}$, and $j \in [J]$:*

$$\left\| \sum_{t=1}^T \mathbb{1}\left[t \in S, a \in \widehat{\mathcal{A}}_t^{c_j,\eta-\mathrm{inf}}\right] (p_t - y_t) \right\|_\infty \le \beta(T^{c_j,S,\eta-\mathrm{inf}}(a))$$

*where $T^{c_j,S,\eta-\mathrm{inf}}(a) = \sum_{t=1}^T \mathbb{1}\left[t \in S, a \in \widehat{\mathcal{A}}_t^{c_j,\eta-\mathrm{inf}}\right]$.*

We will again instantiate UNBIASED-PREDICTION to make predictions that simultaneously achieve decision calibration and infeasibility calibration; we will refer to this instantiation as DECISION-INFEASIBILITY-CALIBRATION-RELAXED. Our guarantees will inherit from the guarantees of UNBIASED-PREDICTION.

**Theorem 4.** *Let $\mathcal{S}$ be a collection of subsequences. Let $\mathcal{N}$ be a set of agents, where each agent is equipped with a utility function $u : \mathcal{A} \times \mathcal{Y} \to [0, 1]$ and $J$ constraint functions $\{c_j : \mathcal{A} \times \mathcal{Y} \to$*

$[-1,1]\}_{j\in[J]}$. *Let $\eta > 0$ be the feasibility tolerance. There is an instantiation of UNBIASED-PREDICTION (Noarov et al., 2023) —which we call DECISION-INFEASIBILITY-CALIBRATION-RELAXED— producing predictions $p_1, ..., p_T \in \mathcal{Y}$ satisfying that for any sequence of outcomes $y_1, ..., y_T \in \mathcal{Y}$, with probability at least $1 - \delta$, for any $(u, \mathbf{c}) \in \mathcal{N}$, $j \in [J]$, $a \in \mathcal{A}$, and $S \in \mathcal{S}$:*

$$\left\| \sum_{t=1}^{T} \mathbb{1}\left[ t \in S, \text{CBR}_{\eta}^{u,\mathbf{c}}(p_t) = a \right] (p_t - y_t) \right\|_{\infty} \leq O\left( \ln(\zeta) \cdot |S|^{1/4} + \sqrt{\ln(\zeta) \cdot T^{u,\mathbf{c},S,\eta}(a)} \right)$$

$$\left\| \sum_{t=1}^{T} \mathbb{1}\left[ t \in S, a \in \widehat{\mathcal{A}}_t^{c_j,\eta-\inf} \right] (p_t - y_t) \right\|_{\infty} \leq O\left( \ln(\zeta) \cdot |S|^{1/4} + \sqrt{\ln(\zeta) \cdot T^{c_j,S,\eta-\inf}(a)} \right)$$

### C.3 THEORETICAL GUARANTEES

Similarly to Lemma 1, we use the infeasibility calibration guarantee to show that for any benchmark action $a \in \mathcal{A}_S^{\mathbf{c},0}$, the number of rounds where it is incorrectly predicted to $\eta$-violate a specific constraint $c_j$ is small. As a result, the number of rounds on which no action is predicted to be $\eta$-feasible is small.

**Lemma 4.** *If the sequence of predictions $p_1, \ldots, p_T$ is $(\mathcal{N}, \mathcal{S}, \eta, \beta)$-infeasibility calibrated, then for any agent $(u, \mathbf{c}) \in \mathcal{N}$, subsequence $S \in \mathcal{S}$, benchmark action $a \in \mathcal{A}_S^{\mathbf{c},0}$, and constraint $j \in [J]$, the number of rounds $T^{c_j,S,\eta-\inf}(a)$ within $S$ on which $a$ is predicted to $\eta$-violate the $j$-th constraint is bounded by:*

$$T^{c_j,S,\eta-\inf}(a) \leq f_\beta(L_\mathcal{C}/\eta)$$

*Consequently, the number of rounds within $S$ on which no actions are predicted to be feasible is bounded by:*

$$\left| \left\{ t \in S : \widehat{\mathcal{A}}_t^{\mathbf{c},\eta-\text{fea}} = \emptyset \right\} \right| \leq J f_\beta(L_\mathcal{C}/\eta)$$

*In particular, plugging in the guarantee from Theorem 4 yields the following concrete form of $f_\beta(L_\mathcal{C}/\eta)$, which holds with probability at least $1 - \delta$:*

$$f_\beta(L_\mathcal{C}/\eta) = O\left( \left( \frac{L_\mathcal{C}|S|^{1/4}}{\eta} + \frac{L_\mathcal{C}^2}{\eta^2} \right) \ln(\zeta) \right)$$

*Proof.* The proof is similar to that of Lemma 1.

On any round $t$ where action $a$ is $\eta$-predicted to violate the $j$-th constraint, we have:

$$c_j(a, p_t) > \eta$$

Since $a$ is in the benchmark class $\mathcal{A}_S^{\mathbf{c},0}$, we have:

$$c_j(a, y_t) \leq 0$$

Combining these two facts gives $c_j(a, p_t) - c_j(a, y_t) > \eta$. Summing this difference over all rounds in $S$ where $a$ is predicted to $\eta$-violate the $j$-th constraint, we get:

$$\sum_{t=1}^{T} \mathbb{1}\left[ t \in S, a \in \widehat{\mathcal{A}}_t^{c_j,\eta-\inf} \right] (c_j(a, p_t) - c_j(a, y_t)) > \eta T^{c_j,S,\eta-\inf}(a)$$

The left-hand side can be bounded using the properties of our predictions. By linearity and $L_\mathcal{C}$-Lipschitzness of the constraint function, and by $(\mathcal{N}, \mathcal{S}, \eta, \beta)$-infeasibility calibration, we have that:

$$\sum_{t=1}^{T} \mathbb{1}\left[ t \in S, a \in \widehat{\mathcal{A}}_t^{c_j,\eta-\inf} \right] (c_j(a, p_t) - c_j(a, y_t))$$

$$= c_j \left( a, \sum_{t=1}^T p_t \mathbb{1}\left[ t \in S, a \in \widehat{\mathcal{A}}_t^{c_j, \eta - \inf} \right] \right) - c_j \left( a, \sum_{t=1}^T y_t \mathbb{1}\left[ t \in S, a \in \widehat{\mathcal{A}}_t^{c_j, \eta - \inf} \right] \right)$$

$$\leq L_{\mathcal{C}} \left\| \sum_{t=1}^T p_t \mathbb{1}\left[ t \in S, a \in \widehat{\mathcal{A}}_t^{c_j, \eta - \inf} \right] - \sum_{t=1}^T y_t \mathbb{1}\left[ t \in S, a \in \widehat{\mathcal{A}}_t^{c_j, \eta - \inf} \right] \right\|_\infty$$

$$\leq L_{\mathcal{C}} \beta(T^{c_j, S, \eta - \inf}(a))$$

Combining these inequalities gives the first part of the lemma:

$$\eta T^{c_j, S, \eta - \inf}(a) < L_{\mathcal{C}} \beta(T^{c_j, S, \eta - \inf}(a))$$

For the concrete bound, we substitute the explicit form for $\beta$ from Theorem 4:

$$\eta f_\beta(L_{\mathcal{C}}/\eta) = L_{\mathcal{C}} O \left( \ln \frac{dJ|\mathcal{A}||\mathcal{N}||\mathcal{S}|T}{\delta} \cdot |S|^{1/4} + \sqrt{\ln \frac{dJ|\mathcal{A}||\mathcal{N}||\mathcal{S}|T}{\delta} \cdot f_\beta(L_{\mathcal{C}}/\eta)} \right)$$

Solve for $f_\beta(L_{\mathcal{C}}/\eta)$ yields the stated form:

$$f_\beta(L_{\mathcal{C}}/\eta) = O \left( \frac{L_{\mathcal{C}}|S|^{1/4} \ln(dJ|\mathcal{A}||\mathcal{N}||\mathcal{S}|T/\delta)}{\eta} + \frac{L_{\mathcal{C}}^2 \ln(dJ|\mathcal{A}||\mathcal{N}||\mathcal{S}|T/\delta)}{\eta^2} \right)$$

Fix any arbitrary benchmark action $a \in \mathcal{A}_S^{\mathbf{c},0}$. If on round $t \in S$, no action is predicted to be feasible, then it must be that $a$ is predicted to be infeasible. Hence, there must exist at least one constraint that $a$ is predicted to violate. As a result, we have:

$$\left| \left\{ t \in S : \widehat{\mathcal{A}}_t^{\mathbf{c}, \eta - \text{fea}} = \emptyset \right\} \right| \leq |\{ t \in S : \exists j \in [J], c_j(a, p_t) > \eta \}|$$

$$\leq \sum_{j=1}^J T^{c_j, S, \eta - \inf}(a)$$

$$\leq J f_\beta(L_{\mathcal{C}}/\eta)$$

$\square$

### C.3.1 BOUNDING THE CUMULATIVE CONSTRAINT VIOLATION

The bound on the cumulative constraint violation follows the same argument as in Theorem 2. The key difference is that the agent's relaxed decision rule, which tolerates violations up to $\eta$ at each round, introduces an additional error term $\eta|S|$.

**Theorem 5.** *Let $\mathcal{S}$ be a collection of subsequences. Let $\mathcal{N}$ be a set of agents, where each agent is equipped with a utility function $u : \mathcal{A} \times \mathcal{Y} \to [0,1]$ and $J$ constraint functions $\{c_j : \mathcal{A} \times \mathcal{Y} \to [-1,1]\}_{j \in [J]}$. Let $\eta > 0$ be the feasibility tolerance. Suppose each agent plays $\eta$-constrained best responses to $p_t$ to compete with actions from the benchmark class $\mathcal{A}_S^{\mathbf{c},0}$ over each subsequence $S \in \mathcal{S}$. If the sequence of predictions $p_1, \ldots, p_T$ is $(\mathcal{N}, \mathcal{S}, \eta, \alpha)$-decision calibrated and $(\mathcal{N}, \mathcal{S}, \eta, \beta)$-infeasibility calibrated, then the cumulative constraint violation of any agent over any subsequence $S \in \mathcal{S}$ is bounded by:*

$$\text{CCV}(S) \leq L_{\mathcal{C}}|\mathcal{A}|\alpha(|S|/|\mathcal{A}|) + J f_\beta(L_{\mathcal{C}}/\eta) + \eta|S|$$

*In particular, plugging in the guarantee from Theorem 4 yields the following bound, which holds with probability at least $1 - \delta$:*

$$\text{CCV}(S) \leq O \left( \left( L_{\mathcal{C}}|\mathcal{A}||S|^{1/4} + L_{\mathcal{C}}\sqrt{|\mathcal{A}||S|} + \frac{JL_{\mathcal{C}}|S|^{1/4}}{\eta} + \frac{JL_{\mathcal{C}}^2}{\eta^2} \right) \ln(\zeta) + \eta|S| \right)$$

By setting $\eta = |T|^{-1/3}$, we arrive at the following concrete bound for the cumulative constraint violation.

**Corollary 5.** *Let $\mathcal{S}$ be a collection of subsequences. Let $\mathcal{N}$ be a set of agents, where each agent is equipped with a utility function $u : \mathcal{A} \times \mathcal{Y} \to [0, 1]$ and $J$ constraint functions $\{c_j : \mathcal{A} \times \mathcal{Y} \to [-1, 1]\}_{j \in [J]}$. Let $\eta > 0$ be the feasibility tolerance. Suppose each agent plays $\eta$-constrained best responses to $p_t$ to compete with actions from the benchmark class $\mathcal{A}_S^{\mathbf{c},0}$ over each subsequence $S \in \mathcal{S}$. The sequence of predictions $p_1, \ldots, p_T$ produced by DECISION-INFEASIBILITY-CALIBRATION guarantees that with probability at least $1 - \delta$, the cumulative constraint violation of any agent over any subsequence $S \in \mathcal{S}$ is bounded by:*

$$\mathrm{CCV}(S) \le O\left(\left(L_{\mathcal{C}}|\mathcal{A}||S|^{1/4} + L_{\mathcal{C}}\sqrt{|\mathcal{A}||S|} + J(L_{\mathcal{C}} + L_{\mathcal{C}}^2)T^{2/3}\right)\ln(\zeta)\right)$$

### C.3.2 BOUNDING THE REGRET

The bound on the constrained swap regret follows the same argument as in Theorem 3.

**Theorem 6.** *Let $\mathcal{S}$ be a collection of subsequences. Let $\lambda : \mathbb{N} \to (0, \infty)$ be a margin function. Let $\mathcal{N}$ be a set of agents, where each agent is equipped with a utility function $u : \mathcal{A} \times \mathcal{Y} \to [0, 1]$ and $J$ constraint functions $\{c_j : \mathcal{A} \times \mathcal{Y} \to [-1, 1]\}_{j \in [J]}$. Let $\eta > 0$ be the feasibility tolerance. Suppose each agent plays $\eta$-constrained best responses to $p_t$ to compete with actions from the benchmark class $\mathcal{A}_S^{\mathbf{c},0}$ over each subsequence $S \in \mathcal{S}$. If the sequence of predictions $p_1, \ldots, p_T$ is $(\mathcal{N}, \mathcal{S}, \eta, \alpha)$-decision calibrated and $(\mathcal{N}, \mathcal{S}, \eta, \beta)$-infeasibility calibrated, then the constrained swap regret of any agent over any subsequence $S \in \mathcal{S}$ is bounded by:*

$$\mathrm{Reg}_{\mathrm{swap}}(u, \mathbf{c}, 0, S) \le 2L_{\mathcal{U}}|\mathcal{A}|\alpha(|S|/|\mathcal{A}|) + J|\mathcal{A}|f_\beta(L_{\mathcal{C}}/\eta)$$

*In particular, plugging in the guarantee from Theorem 4 yields the following bound, which holds with probability at least $1 - \delta$:*

$$\mathrm{Reg}_{\mathrm{swap}}(u, \mathbf{c}, 0, S) \le O\left(\left(L_{\mathcal{U}}|\mathcal{A}||S|^{1/4} + L_{\mathcal{U}}\sqrt{|\mathcal{A}||\mathcal{S}|} + \frac{JL_{\mathcal{C}}|\mathcal{A}||S|^{1/4}}{\eta} + \frac{JL_{\mathcal{C}}^2|\mathcal{A}|}{\eta^2}\right)\ln(\zeta)\right)$$

By setting $\lambda = T^{-1/3}$, we arrive at the following concrete bound for the constrained swap regret.

**Corollary 6.** *Let $\mathcal{S}$ be a collection of subsequences. Let $\mathcal{N}$ be a set of agents, where each agent is equipped with a utility function $u : \mathcal{A} \times \mathcal{Y} \to [0, 1]$ and $J$ constraint functions $\{c_j : \mathcal{A} \times \mathcal{Y} \to [-1, 1]\}_{j \in [J]}$. Let $\eta > 0$ be the feasibility tolerance. Suppose each agent plays $\eta$-constrained best responses to $p_t$ to choose action $a_t$. The sequence of predictions $p_1, \ldots, p_T$ produced by DECISION-INFEASIBILITY-CALIBRATION-RELAXED guarantees that with probability at least $1 - \delta$, the constrained swap regret against the benchmark class $\mathcal{A}_S^{\mathbf{c},0}$ of any agent over any subsequence $S \in \mathcal{S}$ is bounded by:*

$$\mathrm{Reg}_{\mathrm{swap}}(u, \mathbf{c}, 0, S) \le O\left(\left(L_{\mathcal{U}}|\mathcal{A}||S|^{1/4} + L_{\mathcal{U}}\sqrt{|\mathcal{A}||\mathcal{S}|} + J(L_{\mathcal{C}} + L_{\mathcal{C}}^2)|\mathcal{A}|T^{2/3}\right)\ln(\zeta)\right)$$

We then achieve low constrained swap adaptive regret by instantiating our framework with the collection of all contiguous intervals, $\mathcal{S} = \{[t_1, t_2] : 1 \le t_1 \le t_2 \le T\}$.

**Corollary 7.** *Let $\mathcal{N}$ be a set of agents, where each agent is equipped with a utility function $u : \mathcal{A} \times \mathcal{Y} \to [0, 1]$ and $J$ constraint functions $\{c_j : \mathcal{A} \times \mathcal{Y} \to [-1, 1]\}_{j \in [J]}$. Let $\eta > 0$ be the feasibility tolerance. Suppose each agent plays $\eta$-constrained best responses to $p_t$ to choose action $a_t$. The sequence of predictions $p_1, \ldots, p_T$ produced by DECISION-INFEASIBILITY-CALIBRATION-RELAXED guarantees that with probability at least $1 - \delta$, the constrained swap adaptive regret of any agent is bounded by:*

$$\mathrm{Reg}_{\mathrm{swap-adapt}}(u, \mathbf{c}, 0) \le O\left(\left((L_{\mathcal{U}} + JL_{\mathcal{C}} + JL_{\mathcal{C}}^2)|\mathcal{A}|T^{2/3}\right)\ln(\zeta)\right)$$

A dynamic benchmark with $\Delta$ changes partitions the entire time horizon into $\Delta + 1$ intervals. By summing our per-subsequence regret bound over this specific partition, we obtain the following dynamic regret guarantee.

**Corollary 8.** *Let $\mathcal{N}$ be a set of agents, where each agent is equipped with a utility function $u : \mathcal{A} \times \mathcal{Y} \to [0, 1]$ and $J$ constraint functions $\{c_j : \mathcal{A} \times \mathcal{Y} \to [-1, 1]\}_{j \in [J]}$. Let $\eta > 0$ be the feasibility tolerance. Suppose each agent plays $\eta$-constrained best responses to $p_t$ to choose action $a_t$.*

*The sequence of predictions $p_1, \ldots, p_T$ produced by DECISION-INFEASIBILITY-CALIBRATION-RELAXED guarantees that with probability at least $1 - \delta$, the constrained swap dynamic regret of any agent against any piecewise feasible sequence of action modification rule $\vec{\phi} \in (\mathcal{A}^{\mathcal{A}})^T$ is bounded by:*

$$\mathrm{Reg}_{\mathrm{swap-dyn}}(u, \vec{\phi}) \leq O\left(\left((L_{\mathcal{U}} + JL_{\mathcal{C}} + JL_{\mathcal{C}}^2)|\mathcal{A}|T^{2/3}\Delta(\vec{\phi})\right)\ln(\zeta)\right)$$

*A benchmark sequence $\vec{\phi}$ is piecewise feasible if on each interval of constancy $I_k$, the corresponding rule $\psi_k$ maps to the set of actions that are feasible over that entire interval, i.e., $\psi_k : \mathcal{A} \to \mathcal{A}_{I_k}^{\mathbf{c},0}$.*

# D UNBIASED PREDICTION ALGORITHM

In this section we present the UNBIASED-PREDICTION algorithm of Noarov et al. (2023). We first introduce several notations and concepts from Noarov et al. (2023). Let $\Pi = \{(x, p, y) \in \mathcal{X} \times \mathcal{Y} \times \mathcal{Y}\}$ denote the set of possible realized triples at each round. An interaction over $T$ rounds produces a transcript $\pi_T \in \Pi^T$. We write $\pi_T^{<t}$ as the prefix of the first $t - 1$ triples in $\pi_T$, for any $t \leq T$. An *event* $E \in \mathcal{E}$ is a mapping from contexts and predictions to $[0, 1]$, i.e. $E : \mathcal{X} \times \mathcal{Y} \to [0, 1]$.

The UNBIASED-PREDICTION algorithm makes predictions that are unbiased conditional on a collection of events $\mathcal{E}$. The algorithm's conditional bias guarantee depends logarithmically on the number of events:

**Theorem 7.** *(Noarov et al., 2023) For a collection of events $\mathcal{E}$ and convex prediction/outcome space $\mathcal{Y} \subseteq [0, 1]^d$, Algorithm 1 outputs, on any $T$-round transcript $\pi_T$, a sequence of distributions over predictions $\psi_1, ..., \psi_T \in \Delta\mathcal{Y}$ such that for any $E \in \mathcal{E}$:*

$$\left\|\sum_{t=1}^{T} \mathop{\mathbb{E}}_{p_t \sim \psi_t}[E(x_t, p_t)(p_t - y_t)]\right\|_{\infty} \leq O\left(\ln(d|\mathcal{E}|T) + \sqrt{\ln(d|\mathcal{E}|T) \cdot \sum_{t=1}^{T} \mathop{\mathbb{E}}_{p_t \sim \psi_t}[E(x_t, p_t)]}\right).$$

*The algorithm can be implemented with per-round running time scaling polynomially in $d$ and $|\mathcal{E}|$.*

Theorem 7 is stated as expected error bounds over randomized predictions $\psi_t \in \Delta\mathcal{Y}$. In the following Corollary 9, we state the guarantee based on realized predictions $p_t$ that are sampled from $\psi_t$, and generalize it to our multi-subsequence framework. Our guarantees in Theorem 1 directly follow from Corollary 9.

**Corollary 9.** *Let $\mathcal{S}$ be a collection of subsequences. For a collection of events $\mathcal{E}$ and convex prediction/outcome space $\mathcal{Y} \subseteq [0, 1]^d$, Algorithm 1 instantiated with the event collection $\{\mathbb{1}[t \in S] \cdot E\}_{S \in \mathcal{S}, E \in \mathcal{E}}$ outputs, on any $T$-round transcript $\pi_T$, a sequence of predictions $p_1, ..., p_T \in \mathcal{Y}$ satisfying that, with probability at least $1 - \delta$:*

$$\left\|\sum_{t \in S} E(x_t, p_t)(p_t - y_t)\right\|_{\infty} \leq O\left(\ln(d|\mathcal{E}||\mathcal{S}|T/\delta) \cdot |S|^{1/4} + \sqrt{\ln(d|\mathcal{E}||\mathcal{S}|T/\delta) \cdot \sum_{t \in S} E(x_t, p_t)}\right).$$

*The algorithm can be implemented with per-round running time scaling polynomially in $d$, $|\mathcal{E}|$, and $|\mathcal{S}|$.*

*Proof.* Fix any $m \in [d]$, $E \in \mathcal{E}$, and $S \in \mathcal{S}$. Consider the sequence $\{E(x_t, p_t)(p_{t,m} - y_{t,m}) - \mathbb{E}_{p_t \sim \psi_t}[E(x_t, p_t)(p_{t,m} - y_{t,m})]\}_{t=1}^{T}$, where $p_{t,m}$ and $y_{t,m}$ are the $m$-th coordinate of $p_t$ and $y_t$, respectively. It is a sequence of martingale differences, since for any $t \in [T]$:

$$\mathbb{E}\left[E(x_t, p_t)(p_{t,m} - y_{t,m}) - \mathop{\mathbb{E}}_{p_t \sim \psi_t}[E(x_t, p_t)(p_{t,m} - y_{t,m})] \mid \sigma(\pi_T^{<t}, x_t)\right] = 0.$$

The subsequence of these terms corresponding to rounds $t \in S$, i.e., $\{E(x_t, p_t)(p_{t,m} - y_{t,m}) - \mathbb{E}_{p_t \sim \psi_t}[E(x_t, p_t)(p_{t,m} - y_{t,m})]\}_{t \in S}$, is also a martingale difference sequence, because the selection rule is predictable with respect to the filtration $\sigma(\pi_T^{<t}, x_t)$.

By Freedman's inequality (Lemma 7), we have that with probability at least $1 - \frac{\delta}{d|\mathcal{E}||\mathcal{S}|}$:

$$\left|\sum_{t \in S} E(x_t, p_t)(p_{t,m} - y_{t,m}) - \sum_{t \in S} \mathop{\mathbb{E}}_{p_t \sim \psi_t}[E(x_t, p_t)(p_{t,m} - y_{t,m})]\right|$$

$$\leq O\left(\sqrt{\ln(d|\mathcal{E}||\mathcal{S}|\ln(|S|)/\delta) \cdot \sum_{t \in S} \mathbb{E}\left[\left(E(x_t, p_t)(p_{t,m} - y_{t,m}) - \mathop{\mathbb{E}}_{p_t \sim \psi_t}[E(x_t, p_t)(p_{t,m} - y_{t,m})]\right)^2 \mid \sigma(\pi_T^{<t}, x_t)\right]}\right.$$

$$\left. + \ln(d|\mathcal{E}||\mathcal{S}|\ln(|S|)/\delta)\right)$$

$$\leq O\left(\sqrt{\ln(d|\mathcal{E}||\mathcal{S}|/\delta) \cdot \sum_{t \in S} \mathbb{E}\left[(E(x_t, p_t)(p_{t,m} - y_{t,m}))^2 \mid \sigma(\pi_T^{\leq t}, x_t)\right]} + \ln(d|\mathcal{E}||\mathcal{S}|/\delta)\right)$$

$$\leq O\left(\sqrt{\ln(d|\mathcal{E}||\mathcal{S}|\ln(|S|)/\delta) \cdot \sum_{t \in S} \mathbb{E}\left[E(x_t, p_t) \mid \sigma(\pi_T^{\leq t}, x_t)\right]} + \ln(d|\mathcal{E}||\mathcal{S}|\ln(|S|)/\delta)\right)$$

where the second inequality follows from the fact that the conditional variance is less than or equal to the conditional second moment, and the third inequality follows from the fact that $E(x_t, p_t) \in [0, 1]$ and $|p_{t,m} - y_{t,m}| \in [0, 1]$.

Using the union bound over all $m \in [d]$, $E \in \mathcal{E}$, and $S \in \mathcal{S}$, we have that with probability at least $1 - \delta$, for any $E \in \mathcal{E}$ and any $S \in \mathcal{S}$:

$$\left\|\sum_{t \in S} E(x_t, p_t)(p_t - y_t) - \sum_{t \in S} \mathop{\mathbb{E}}_{p_t \sim \psi_t}[E(x_t, p_t)(p_t - y_t)]\right\|_\infty$$

$$\leq O\left(\sqrt{\ln(d|\mathcal{E}||\mathcal{S}|\ln(|S|)/\delta) \cdot \sum_{t \in S} \mathbb{E}\left[E(x_t, p_t) \mid \sigma(\pi_T^{\leq t}, x_t)\right]} + \ln(d|\mathcal{E}||\mathcal{S}|\ln(|S|)/\delta)\right)$$

By applying Theorem 7 with the event collection $\{\mathbb{1}[t \in S] \cdot E\}_{S \in \mathcal{S}, E \in \mathcal{E}}$ and combining its guarantee with the deviation bound above, we derive that with probability at least $1 - \delta$, for any $E \in \mathcal{E}$ and any $S \in \mathcal{S}$:

$$\left\|\sum_{t \in S} E(x_t, p_t)(p_t - y_t)\right\|_\infty$$

$$\leq \left\|\sum_{t=1}^{T} \mathop{\mathbb{E}}_{p_t \sim \psi_t}[\mathbb{1}[t \in S] E(x_t, p_t)(p_t - y_t)]\right\|_\infty$$

$$+ \left\|\sum_{t \in S} E(x_t, p_t)(p_t - y_t) - \sum_{t \in S} \mathop{\mathbb{E}}_{p_t \sim \psi_t}[E(x_t, p_t)(p_t - y_t)]\right\|_\infty$$

$$\leq O\left(\ln(d|\mathcal{E}||S|T) + \sqrt{\ln(d|\mathcal{E}||S|T) \cdot \sum_{t \in S} \mathop{\mathbb{E}}_{p_t \sim \psi_t}[E(x_t, p_t)]}\right)$$

$$+ O\left(\sqrt{\ln(d|\mathcal{E}||\mathcal{S}|\ln(|S|)/\delta) \cdot \sum_{t \in S} \mathbb{E}\left[E(x_t, p_t) \mid \sigma(\pi_T^{\leq t}, x_t)\right]} + \ln(d|\mathcal{E}||\mathcal{S}|\ln(|S|)/\delta)\right)$$

$$\leq O\left(\sqrt{\ln(d|\mathcal{E}||S|T/\delta) \cdot \sum_{t \in S} \mathop{\mathbb{E}}_{p_t \sim \psi_t}[E(x_t, p_t)]}\right.$$

$$\left. + \sqrt{\ln(d|\mathcal{E}||\mathcal{S}|T/\delta) \cdot \sum_{t \in S} \mathbb{E}\left[E(x_t, p_t) \mid \sigma(\pi_T^{\leq t}, x_t)\right]} + \ln(d|\mathcal{E}||\mathcal{S}|T/\delta)\right).$$

The next step is to bound the deviation of $\sum_{t \in S} E(x_t, p_t)$ from its expectation $\sum_{t \in S} \mathbb{E}_{p_t \sim \psi_t}[E(x_t, p_t)]$ and from its conditional-expectation-based version $\sum_{t \in S} \mathbb{E}\left[E(x_t, p_t) \mid \sigma(\pi_T^{\leq t}, x_t)\right]$ for any $S \in \mathcal{S}$ and $E \in \mathcal{E}$. We will again apply a martingale concentration inequality.

Fix any $E \in \mathcal{E}$ and $S \in \mathcal{S}$. Consider the sequence $\{E(x_t, p_t) - \mathbb{E}_{p_t \sim \psi_t}[E(x_t, p_t)]\}_{t=1}^{T}$ and the sequence $\{E(x_t, p_t) - \mathbb{E}\left[E(x_t, p_t) \mid \sigma(\pi_T^{<t}, x_t)\right]\}_{t=1}^{T}$. Both of them are sequences of martingale differences, since for any $t \in [T]$:

$$\mathbb{E}\left[E(x_t, p_t) - \mathop{\mathbb{E}}_{p_t \sim \psi_t}[E(x_t, p_t)] \mid \sigma(\pi_T^{<t}, x_t)\right] = 0$$

$$\mathbb{E}\left[E(x_t, p_t) - \mathbb{E}\left[E(x_t, p_t) \mid \sigma(\pi_T^{<t}, x_t)\right] \mid \sigma(\pi_T^{<t}, x_t)\right] = 0$$

The subsequence of these terms corresponding to rounds $t \in S$, i.e., $\{E(x_t, p_t) - \mathbb{E}_{p_t \sim \psi_t}[E(x_t, p_t)]\}_{t \in S}$ and $\{E(x_t, p_t) - \mathbb{E}\left[E(x_t, p_t) \mid \sigma(\pi_T^{<t}, x_t)\right]\}_{t \in S}$, are also both martingale difference sequences, because the selection rule is predictable with respect to the filtration $\sigma(\pi_T^{<t}, x_t)$.

By Azuma-Hoeffding inequality (Lemma 5), we have that with probability at least $1 - \frac{\delta}{2d|\mathcal{E}||\mathcal{S}|}$:

$$\left|\sum_{t \in S} E(x_t, p_t) - \sum_{t \in S} \mathop{\mathbb{E}}_{p_t \sim \psi_t}[E(x_t, p_t)]\right| \leq 2\sqrt{2\ln(4d|\mathcal{E}||\mathcal{S}|/\delta) \cdot |S|}$$

$$\left|\sum_{t \in S} E(x_t, p_t) - \sum_{t \in S} \mathbb{E}\left[E(x_t, p_t) \mid \sigma(\pi_T^{<t}, x_t)\right]\right| \leq 2\sqrt{2\ln(4d|\mathcal{E}||\mathcal{S}|/\delta) \cdot |S|}.$$

Using the union bound over all $E \in \mathcal{E}$ and $S \in \mathcal{S}$, we have that with probability at least $1 - \delta$, for any $E \in \mathcal{E}$ and any $S \in \mathcal{S}$:

$$\left|\sum_{t \in S} E(x_t, p_t) - \sum_{t \in S} \mathop{\mathbb{E}}_{p_t \sim \psi_t}[E(x_t, p_t)]\right| \leq O\left(\sqrt{\ln(d|\mathcal{E}||\mathcal{S}|/\delta) \cdot |S|}\right)$$

$$\left|\sum_{t \in S} E(x_t, p_t) - \sum_{t \in S} \mathbb{E}\left[E(x_t, p_t) \mid \sigma(\pi_T^{<t}, x_t)\right]\right| \leq O\left(\sqrt{\ln(d|\mathcal{E}||\mathcal{S}|/\delta) \cdot |S|}\right)$$

Finally, substituting the above concentration bound for $\sum_{t \in S} E(x_t, p_t)$ into our high-probability guarantee yields the final result, with probability at least $1 - \delta$.

$$\left\|\sum_{t \in S} E(x_t, p_t)(p_t - y_t)\right\|_{\infty}$$

$$\leq O\left(\sqrt{\ln(d|\mathcal{E}||S|T/\delta) \cdot \sum_{t \in S}\left(E(x_t, p_t) + O\left(\sqrt{\ln(d|\mathcal{E}||\mathcal{S}|/\delta) \cdot |S|}\right)\right)} + \ln(d|\mathcal{E}||\mathcal{S}|T/\delta)\right)$$

$$\leq O\left(\ln(d|\mathcal{E}||\mathcal{S}|T/\delta) \cdot |S|^{1/4} + \sqrt{\ln(d|\mathcal{E}||S|T/\delta) \cdot \sum_{t \in S} E(x_t, p_t)}\right)$$

$\square$

# E    HANDLING THE EXPECTED-CONSTRAINT BENCHMARK

In our main text, we focused on the *realized-constraint benchmark* $\mathcal{A}_S^{\mathbf{c},\lambda}$, where actions must satisfy $c_j(a, y_t) \leq -\lambda$ in every round $t \in S$. Prior work Bechavod et al. (2025) also considered the *expected-constraint benchmark* $\mathcal{A}_S^{\mathbb{E}[\mathbf{c}],\lambda}$, where actions need only satisfy $\mathbb{E}_{y_t \sim Y_t}[c_j(a, y_t)] \leq -\lambda$. In Bechavod et al. (2025), handling this benchmark was more involved and required a different algorithm than the one used for the realized-constraint benchmark.

Here, we show that our approach is general enough to handle $\mathcal{A}_S^{\mathbb{E}[\mathbf{c}],\lambda}$ with the *same algorithm* and achieve the *same asymptotic bounds*. The core of our guarantees (e.g., Theorem 2 and Theorem 3) relies on Lemma 1, which bounds the number of times a benchmark action is incorrectly predicted

as infeasible. We now show how the proof of Lemma 1 can be adapted for the expected-constraint benchmark.

Let $a \in \mathcal{A}_S^{\mathbb{E}[\mathbf{c}],\lambda}$ be a benchmark action from the expected-constraint class. The proof proceeds analogously to that of Lemma 1. On any round $t$ where action $a$ is predicted to violate the $j$-th constraint, we have $c_j(a, p_t) > 0$.

The next step of the original proof, however, no longer holds. We cannot claim $c_j(a, y_t) \leq -\lambda$. Instead, for the expected-constraint benchmark, we only know that $\mathbb{E}_{y_t \sim Y_t}[c_j(a, y_t)] \leq -\lambda$. Combining these two facts gives a different gap: $c_j(a, p_t) - \mathbb{E}_{y_t \sim Y_t}[c_j(a, y_t)] > \lambda$. Summing this difference over all rounds in $S$ where $a$ is predicted to violate the $j$-th constraint, we get:

$$\sum_{t=1}^{T} \mathbb{1}\left[t \in S, a \in \widehat{\mathcal{A}}_t^{c_j,\text{inf}}\right] \left(c_j(a, p_t) - \mathbb{E}_{y_t \sim Y_t}[c_j(a, y_t)]\right) > \lambda T^{c_j,S,\text{inf}}(a).$$

To connect this to our calibration guarantee, we add and subtract $c_j(a, y_t)$ inside the summation:

$$\lambda T^{c_j,S,\text{inf}}(a) < \sum_{t=1}^{T} \mathbb{1}\left[t \in S, a \in \widehat{\mathcal{A}}_t^{c_j,\text{inf}}\right] \left(c_j(a, p_t) - c_j(a, y_t)\right)$$

$$+ \sum_{t=1}^{T} \mathbb{1}\left[t \in S, a \in \widehat{\mathcal{A}}_t^{c_j,\text{inf}}\right] \left(c_j(a, y_t) - \mathbb{E}_{y_t \sim Y_t}[c_j(a, y_t)]\right).$$

The first term on the right-hand side is bounded exactly as in the original proof of Lemma 1. By linearity, $L_{\mathcal{C}}$-Lipschitzness, and $(\mathcal{N}, \mathcal{S}, \beta)$-infeasibility calibration, we have:

$$\sum_{t=1}^{T} \mathbb{1}\left[t \in S, a \in \widehat{\mathcal{A}}_t^{c_j,\text{inf}}\right] \left(c_j(a, p_t) - c_j(a, y_t)\right) \leq L_{\mathcal{C}}\beta(T^{c_j,S,\text{inf}}(a)).$$

The second term, $\sum_{t=1}^{T} \mathbb{1}\left[t \in S, a \in \widehat{\mathcal{A}}_t^{c_j,\text{inf}}\right] \left(c_j(a, y_t) - \mathbb{E}_{y_t \sim Y_t}[c_j(a, y_t)]\right)$, is new. By the Azuma-Hoeffding inequality (Lemma 5) and a union bound over all $J|\mathcal{A}||\mathcal{S}|$ possible events (accounted for in our $\ln(\zeta)$ term), we have with probability at least $1 - \delta$:

$$\left|\sum_{t=1}^{T} \mathbb{1}\left[t \in S, a \in \widehat{\mathcal{A}}_t^{c_j,\text{inf}}\right] \left(c_j(a, y_t) - \mathbb{E}_{y_t \sim Y_t}[c_j(a, y_t)]\right)\right| \leq O\left(\sqrt{T^{c_j,S,\text{inf}}(a) \ln(\zeta)}\right).$$

Combining these bounds, we get:

$$\lambda T^{c_j,S,\text{inf}}(a) < L_{\mathcal{C}}\beta(T^{c_j,S,\text{inf}}(a)) + O\left(\sqrt{T^{c_j,S,\text{inf}}(a) \ln(\zeta)}\right).$$

Substituting the concrete form of $\beta(\cdot)$ from Theorem 1:

$$\lambda T^{c_j,S,\text{inf}}(a) < L_{\mathcal{C}} \cdot O\left(\ln(\zeta) \cdot |S|^{1/4} + \sqrt{\ln(\zeta) \cdot T^{c_j,S,\text{inf}}(a)}\right) + O\left(\sqrt{T^{c_j,S,\text{inf}}(a) \ln(\zeta)}\right).$$

The new term is asymptotically absorbed by the $O(\sqrt{\ln(\zeta) \cdot T^{c_j,S,\text{inf}}(a)})$ term already present in the calibration guarantee. Thus, the inequality remains asymptotically unchanged:

$$\lambda T^{c_j,S,\text{inf}}(a) < O\left(\ln(\zeta)|S|^{1/4} + \sqrt{\ln(\zeta)T^{c_j,S,\text{inf}}(a)}\right).$$

Solving for $T^{c_j,S,\text{inf}}(a)$ yields the same asymptotic bound as in Lemma 1. Consequently, all subsequent guarantees in Theorem 2 and Theorem 3 that depend on this term hold for the expected-constraint benchmark with the same order of bounds.

## F    Achieving Best-in-Class Predictive Performance

In Section 2, we noted that our framework can incorporate benchmark predictive models. This "best-in-class" guarantee is a powerful feature of the underlying Unbiased-Prediction algorithm from Noarov et al. (2023), and it can be seamlessly added to our approach. The core idea is to augment the collection of conditioning events used by our algorithm. In addition to the events required for decision calibration (Definition 6) and infeasibility calibration (Definition 7), we can add a new set of events designed to compare our predictor $p_t$ against a benchmark class of predictors, $\mathcal{Q}$.

Let $\mathcal{Q}$ be a collection of predictors. Each $q \in \mathcal{Q}$ is a model that maps the feature $x_t$ to a prediction $q(x_t) \in \mathcal{Y}$ of the outcome $y_t$. Following Noarov et al. (2023), we can force our predictor to be "as good as" $q$ by defining events based on the joint values of $p_t$ and $q(x_t)$. This is typically done by discretizing the prediction space into a set of $\delta$-level sets (or bins). For each coordinate $i \in [d]$, predictor $q \in \mathcal{Q}$, and pair of bins $(\Omega_a, \Omega_b)$, we define a new event:

$$E_q^{i,a,b}(p_t) = \mathbb{1}[p_{t,i} \in \Omega_a, q(x_t)_i \in \Omega_b].$$

We then instantiate the Unbiased-Prediction algorithm (which we call Decision-Infeasibility-Calibration in our main text) with the union of our original calibration events and this new collection of model-comparison events.

Running our algorithm on this final, larger set of events yields two guarantees simultaneously. First, since our original calibration events are still included in the collection, all our guarantees for dynamic regret and constraint violation continue to hold. The only change is that the $\ln(\zeta)$ term in our bounds will be slightly larger, as $\zeta$ now depends on the size of this larger event collection, but this is only a logarithmic increase. Second, as proven in Noarov et al. (2023), forcing unbiasedness on these new events guarantees that our predictor $p_t$ achieves a cumulative loss measuring predictive accuracy that is as low as the loss of the best predictor $q \in \mathcal{Q}$, up to a sublinear $o(T)$ term.

In summary, this "stacking" of event sets allows our single set of predictions $p_t$ to simultaneously provide guarantees on downstream decisions (low regret and constraint violation) and on its own predictive accuracy (no worse than the best benchmark model). For a formal proof of the best-in-class guarantee, we refer the reader to Noarov et al. (2023).

## G    Azuma-Hoeffding's Inequality

**Lemma 5.** *Let $Z_1, \ldots, Z_T$ be a martingale difference sequence. $|Z_t| \leq M$ for all t. Then with probability at least $1 - \delta$:*

$$\left| \sum_{t=1}^{T} Z_t \right| \leq M \sqrt{2T \ln \frac{2}{\delta}}$$

## H    Freedman's Inequality

The following lemma gives a standard form of Freedman's inequality, which can be found in works such as Tropp (2011).

**Lemma 6.** *Let $(\mathcal{F}_i)_{i=0}^n$ be a filtration. Let $Z_1, \ldots, Z_n$ be a martingale difference sequence with respect to $(\mathcal{F}_i)_{i=0}^n$. $Z_i \leq M$ for all i. Let $V_n = \sum_{i=1}^n \mathbb{E}[Z_i^2 | \mathcal{F}_{i-1}]$. Then, for all $\tau \geq 0$ and $v > 0$,*

$$P \left( \sum_{i=1}^{n} Z_i \geq \tau, V_n \leq v \right) \leq \exp \left( -\frac{\tau^2/2}{v + M\tau/3} \right)$$

We derive the following convenient form that we use in our proofs.

**Lemma 7.** *Let $(\mathcal{F}_i)_{i=0}^n$ be a filtration. Let $Z_1, \ldots, Z_n$ be a martingale difference sequence with respect to $(\mathcal{F}_i)_{i=0}^n$. $Z_i \leq M$ for all i. Let $V_n = \sum_{i=1}^n \mathbb{E}[Z_i^2 | \mathcal{F}_{i-1}]$. For any $\delta \in (0,1)$, with probability at least $1 - \delta$:*

$$\sum_{i=1}^{n} Z_i \leq 2\sqrt{V_n \left(\ln(1/\delta) + C_{n,M}\right)} + \left(\frac{2}{3}M + 3\right) \left(\ln(1/\delta) + C_{n,M}\right)$$

*where $C_{n,M} = 2\ln(\ln(nM^2) + 1) + 2\ln(2)$.*

*Proof.* Let $\tau = \sqrt{2v\ln(1/\delta)} + \frac{2}{3}M\ln(1/\delta)$, it satisfies that $\frac{\tau^2/2}{v+M\tau/3} > \ln(1/\delta)$, because:

$$
\begin{aligned}
\tau^2/2 - (v + M\tau/3)\ln(1/\delta) &= v\ln(1/\delta) + \frac{2}{9}M^2\ln^2(1/\delta) + \sqrt{2v\ln(1/\delta)} \cdot \frac{2}{3}M\ln(1/\delta) \\
&\quad - v\ln(1/\delta) - \frac{1}{3}M\left(\sqrt{2v\ln(1/\delta)} + \frac{2}{3}M\ln(1/\delta)\right)\ln(1/\delta) \\
&= \sqrt{2v\ln(1/\delta)} \cdot \frac{2}{3}M\ln(1/\delta) - \frac{1}{3}M \cdot \sqrt{2v\ln(1/\delta)} \cdot \ln(1/\delta) \\
&= \frac{1}{3}M \cdot \sqrt{2v\ln(1/\delta)} \cdot \ln(1/\delta) \\
&> 0
\end{aligned}
$$

Applying Lemma 6 with this choice of $\tau$, we derive that for any $\delta \in (0,1)$ and $v > 0$:

$$
P\left(\sum_{i=1}^n Z_i \geq \sqrt{2v\ln(1/\delta)} + \frac{2}{3}M\ln(1/\delta), \ V_n \leq v\right) \leq \delta
$$

Let $K = \lceil\log_2(nM^2)\rceil$. Let $\delta_k = \frac{6\delta}{\pi^2 k^2}$ and $v_k = 2^k$ for $k = 1, \ldots, K$. Instantiate the above inequality with $\delta_k$ and $v_k$, we have for any $k \in [K]$:

$$
P(E_k) \leq \delta_k
$$

where $E_k$ denotes the event:

$$
E_k = \left\{\sum_{i=1}^n Z_i \geq \sqrt{2v_k\ln(1/\delta_k)} + \frac{2}{3}M\ln(1/\delta_k), \ V_n \leq v_k\right\}
$$

Applying the union bound, we have:

$$
\begin{aligned}
P\left(\cup_{k=1}^K E_k\right) &\leq \sum_{k=1}^K P(E_k) \\
&\leq \sum_{k=1}^K \delta_k \\
&= \sum_{k=1}^K \frac{6\delta}{\pi^2 k^2} \\
&< \delta
\end{aligned}
$$

Therefore, with probability at least $1 - \delta$, none of the events $E_k$ occurs. We will prove that our desired guarantee holds true conditional on this high-probability event $\cap_{k=1}^K E_k^c$.

For convenience, let $v_0 = 0$. Then $\{(v_{k-1}, v_k]\}_{k=1}^K$ forms a partition of $(0, 2^K]$. Since $0 < V_n \leq nM^2 \leq 2^K$, there must exist $k \in [1, K]$, such that $v_{k-1} < V_n \leq v_k$.

Since $E_k$ does not happen, it must be that $\sum_{i=1}^n Z_i < \sqrt{2v_k\ln(1/\delta_k)} + \frac{2}{3}M\ln(1/\delta_k)$.

If $k \in [2, K]$, then $v_k = 2v_{k-1} < 2V_n$. Hence, $\sum_{i=1}^n Z_i < \sqrt{4V_n\ln(1/\delta_k)} + \frac{2}{3}M\ln(1/\delta_k)$.

If $k = 1$, then $v_k = 2^k = 2$. Hence, $\sum_{i=1}^n Z_i < 2\sqrt{\ln(1/\delta_k)} + \frac{2}{3}M\ln(1/\delta_k)$.

Therefore, conditional on the event $\cap_{k=1}^K E_k^c$, which happens with probability at least $1 - \delta$, it always holds true that:

$$
\sum_{i=1}^n Z_i < 2\sqrt{V_n\ln(1/\delta_k)} + 2\sqrt{\ln(1/\delta_k)} + \frac{2}{3}M\ln(1/\delta_k)
$$

By definition of $\delta_k$, we have $\ln(1/\delta_k) = \ln \frac{\pi^2 k^2}{6\delta} \geq \ln \frac{\pi^2}{6} > 0.49$. Hence, $\sqrt{\ln(1/\delta_k)} > 0.7$. Then we have:

$$\sum_{i=1}^n Z_i < 2\sqrt{V_n \ln(1/\delta_k)} + 3 \cdot 0.7 \cdot \sqrt{\ln(1/\delta_k)} + \frac{2}{3} M \ln(1/\delta_k)$$

$$< 2\sqrt{V_n \ln(1/\delta_k)} + \left(\frac{2}{3} M + 3\right) \ln(1/\delta_k)$$

We note that

$$\ln(1/\delta_k) = \ln \frac{\pi^2 k^2}{6\delta}$$
$$= 2\ln \frac{\pi k}{\sqrt{6}} + \ln(1/\delta)$$
$$\leq 2\ln \frac{\pi K}{\sqrt{6}} + \ln(1/\delta)$$
$$\leq 2\ln \frac{\pi(\log_2(nM^2) + 1)}{\sqrt{6}} + \ln(1/\delta)$$
$$\leq 2\ln(2\ln(nM^2) + 2) + \ln(1/\delta)$$
$$= C_{n,M} + \ln(1/\delta)$$

Therefore, with probability at least $1 - \delta$:

$$\sum_{i=1}^n Z_i < 2\sqrt{V_n \left(\ln(1/\delta) + C_{n,M}\right)} + \left(\frac{2}{3} M + 3\right) \left(\ln(1/\delta) + C_{n,M}\right)$$

$\square$

# I    USE OF LARGE LANGUAGE MODELS

We used Gemini 2.5 Pro as a writing assistant. Its use was focused on improving language and readability, including tasks such as correcting grammar, refining sentence structure, and ensuring stylistic consistency.

