# OpenReview forum: "Dynamic Regret Bounds for Online Omniprediction with Long Term Constraints"
_ICLR.cc/2026/Conference — Submitted to ICLR 2026_

### Official Review · Reviewer_ivPL · 2025-10-19

**Soundness:** 3
**Presentation:** 2
**Contribution:** 2
**Rating:** 4
**Confidence:** 3

**Summary:**

This paper studies a recently introduced setting in the field of online learning with unknown constraints, that is, online omniprediction with long term constraints. In such a setting, for every round $t\in[T]$, an adversary selects a feature vector and a distribution over an outcome space. Then, a forecaster (namely, the learner) observes at each round the adversarially generated feature vector $x_t$ and then outputs a prediction $p_t$ on the outcome space (notice that, both the utility and costs of the environment at time $t$ depend on the real outcome). Then, the prediction $p_t$ is observed by $\mathcal{N}$ agents which select an action accordingly. The goal is to minimise the violation and the regret (in this paper, not only fixed benchmarks are studied, but also dynamic ones) over the $T$ rounds for all the agents. The authors propose an algorithm for the forecaster, which employs existing techniques in order to output  ”unbiased” predictions for the agents. Thus, the agents select the action optimising greedily on the per-round reward and constraints computed given $p_t$. The algorithm attains sublinear regret and violation which depends on the various benchmarks employed.

**Strengths:**

I believe that this work may hide some interesting results on learning against dynamic benchmarks which can go beyond the setting proposed in the paper. Nonetheless, the presentation does not allow to clearly recognise them. Indeed, there are several weaknesses that worsen the impact of the work.

**Weaknesses:**

I have many concerns on both the presentation and the significance of the results presented in the work. Specifically,

1. It is not clear to me which are the technical contributions of this work. Given the current presentation of the work, the main contribution lies in extending Behaved et al. (2025) to dynamic benchmarks. It is not clearly discussed how the techniques of the two papers differ. Moreover, from a technical perspective, it seems to me that the algorithm of the forecaster is an adaptation of Noarov et al. (2023), while the agents simply optimises greedily. Thus, I do not see any particular algorithmic novelty and the fact that the algorithm is deferred to the Appendix does not help in this sense.
 2. Similarly, the authors could better highlight whether the bounds obtained against dynamic benchmark have been already provided for the standard setting of online learning with unknown constraints or not.
 3. The current version of both the introduction and the preliminaries makes the setting not particularly easy to understand, even for readers familiar with the online learning with constraints literature. For instance, if I understood properly, the paper focuses on discrete action spaces. But as far as I see, it is specified only at line 56. Similarly, I assume that the feature space is somehow related to the outcome one, since the forecaster has to predict over outcomes observing first a feature vector. Nonetheless, I do not see any discussion on this in the preliminaries.
 4. I have some concerns on the setting itself. To me, the main difficulty of this setting is to predict a good $p_t$ (or $\pi_t$) at each round, since both the per round utility and costs of the agents are directly and deterministically estimated given $p_t$. Thus, I do not understand how the existence of unknown constraints should make this particular setting more interesting or challenging. The goal would be predicting a good estimate of the per round outcome even in unconstrained settings.
 5. Another concern is about the benchmarks employed in the work. It seems to me that many of the bounds are computed with respect to solutions which are “strictly feasible at each round”. In footnote 2 the authors state that it is standard to assume Slater’s condition in the literature. I agree with that, but assuming that a strictly feasible solution exists (as in Slater) is very different from using strictly feasible solutions as benchmarks in the regret.
 6. Finally, as a minor comment, the presentation could be improved in many ways throughout the paper. For instance, the equations clearly go beyond the template margins.

**Questions:**

Please see weaknesses. I will be happy to reconsider my score if my concerns are properly tackled.

---

> ### Author Response · Authors · 2025-11-21
> **Response to reviewer**
>
> Thank you for the valuable feedback and questions! We are happy to hear you have found the results in the paper interesting. We have incorporated your suggestions regarding the presentation of the paper in the revision.
>
> Below are answers to your questions:
>
> **It is not clearly discussed how the techniques of in the paper differ from the ones in Bechavod et al. 2025**
>
> First, we point the reviewer to the paragraphs in lines 114 and 131 where we discuss technical implementation differences between the two papers, as well as the differences in resulting bounds. The answer is that yes, the two papers employ very different strategies: Bechavod et al. 2025 relies on an elimination-based algorithm, where each downstream decision maker has to maintain an ``active set'' of actions consistent with incoming constraints. Apart from the additional burden on agents to constantly maintain a state, their implementation also results in bounds that scale linearly with the number of subsequences over which the learner wishes the guarantees to hold. This is not just a quantitative issue --- it makes it impossible to obtain dynamic regret bounds using their machinery. The algorithm we propose in our work, on the other hand, does not require agents to maintain a state, but rather simply asks them to perform simple per-round optimization. The different technique here results in bounds that instead depend only logarithmically with the number of subsequences, allowing us to derive meaningful dynamic regret bounds.
>
> **It seems that the algorithm of the forecaster is an adaptation of Noarov et al. (2023), while the agents simply optimises greedily**
>
> We refer the reviewer to our response to Reviewer Djoj which addresses this question in detail.
>
> **Whether the bounds obtained against dynamic benchmark have been already provided for the standard setting of online learning with unknown constraints or not**
>
> Great question. In the paper we study a stronger variant we term swap-dynamic regret, which has not been studied by any of the past works on online learning with unknown constraints. Our results also pertain to multiple downstream decision makers rather than the special case of a single decision maker as in the standard setting of online learning with unknown constraints. We elaborate on both points in our response to Reviewer 59re.
>
> **The relation between the feature and the outcome spaces**
>
> Good question! Our framework makes no assumptions about the relationship between the feature $x_t$ and the outcome distribution $Y_t$. The adversary can select both $x_t$ and $Y_t$ in an arbitrary manner at the start of each round. Our algorithm is designed to provide guarantees whether $x_t$ is highly informative or completely uninformative about $Y_t$. Furthermore, if one has access to a class of predictive models that predict the outcomes using the features, our framework can be configured to incorporate them and produce predictions that perform at least as well as the best model in that class. We discuss this extension in Appendix F.
>
> **To me, the main difficulty of this setting is to predict a good $p_t$
> (or $\pi_t$) at each round, since both the per round utility and costs of the agents are directly and deterministically estimated given $p_t$. Thus, I do not understand how the existence of unknown constraints should make this particular setting more interesting or challenging. The goal would be predicting a good estimate of the per round outcome even in unconstrained settings.**
>
> It is true that if one were able to produce perfect pointwise predictions of the outcome, then these would be sufficient statistics both for regret minimization and constraint satisfaction. The difficulty is that perfect pointwise prediction is not possible. Instead, we need to produce predictions such that when decision makers *treat them* as if they were perfect pointwise predictions and optimize accordingly, they obtain strong regret and constraint violation bounds, despite their imperfection. It might be useful here to observe that the problem we solve here is only more general/more difficult than the standard single decision maker problem of learning with long-term constraints. Adding long term constraints to the single decision makers problem makes the problem more difficult (compared to just regret minimization) for the same reason it does so here; we show that the problem can be overcome even in the omniprediction variant where we need to produce a single stream of predictions to simultaniously feed many downstream decision makers with differing goals.

---

> > ### Author Response · Authors · 2025-11-21
> > **Response to reviewer (continued)**
> >
> > **It seems to me that many of the bounds are computed with respect to solutions which are ``strictly feasible at each round''. There is a difference between Slater's condition and strict per-round feasibility.**
> >
> > We note that in the paper we provide results even when dispensing with the assumption of strong feasibility, as we mention in Remark 4. Regardless, the reviewer is correct to point out the difference between Slater's condition and the benchmark that is strongly feasible. We do  not use the two interchangeably, and we added a clarifying comment about this in the revision.

---

### Official Review · Reviewer_59re · 2025-10-29

**Soundness:** 3
**Presentation:** 3
**Contribution:** 1
**Rating:** 2
**Confidence:** 3

**Summary:**

This paper studies a version of the online omniprediction problem, which is an online learning problem where the learner produces a prediction of the adversary's action in each round and an agent uses this prediction to choose an action. In particular, they consider the version of this problem where there is both a utility function and a constraint function. The goal is to guarantee sublinear regret and cumulative constraint violation.

**Strengths:**

- The paper presents a well-defined framework for studying the online omni prediction under constraints.

**Weaknesses:**

1. To my understanding, this setting fits under the problem of constrained online convex optimization (COCO). In particular, the decision rule of the agent (Definition 5) can just be viewed as a computation Oracle within the COCO framework. For example, this could be taken to be quadratic minimization that is frequently used in the COCO setting (i.e. as in Neely and Yu (2017) and following works). In that sense, the proposed framework in the paper can be seen as COCO with a constrained minimization Oracle. However, it is not clear that the paper makes substantial contribution to the COCO literature for the following reasons:
   - The $O(\sqrt{T})$ regret and CCV bounds in the paper are only against benchmarks that are $\lambda$ strictly-feasible (i.e. $a : c(a) \leq -\lambda$). This is a stronger assumption than the typical one to get $O(\sqrt{T})$ regret and CCV bounds in the COCO literature: the existence of a $\lambda$ strictly-feasible point. This is widely known in the COCO literature since Neely and Yu (2017).
   - There does not appear to be computational benefits of the proposed agent decision rule/computation oracle over existing ones in the literature (e.g. unconstrained quadratic minimization).
   - The setting is restricted to the case of linear utility and constraint functions.
2. The paper mentioned that the regret notion it considers is stronger than the dynamic regret considered by prior work in COCO. However, it is unclear how this is the case. In particular, it seems that they consider a very specific variation notion being the number of times that the benchmark sequence changes. I did not see any discussion about how this variation notion relates to existing ones in the COCO dynamic regret literature.

**Questions:**

1. How does the proposed variation notion in the dynamic regret relates to those studied in the literature? For example, as in Guo et al. (2022) or Liu et al. (2022).
2. What is the order-wise dynamic regret bound in the paper? How does this compare with the literature?

---

> ### Author Response · Authors · 2025-11-21
> **Response to reviewer**
>
> Thank you for the valuable feedback and questions!
>
> Below are answers to your questions:
>
> **The setting of the paper fits under the problem of constrained online convex optimization (COCO)**
>
> No! The key novelty of our work relative to the large body of work on constrained online convex optimization is that we are solving the more general problem of providing a single stream of forecasts that can be naively/straightforwardly consumed by many downstream decision makers, each with their own objective and constraints, and offer each of them both regret and constraint violation bounds of the sort that they could have gotten by running algorithms from the COCO literature. The key and important difference is that the downstream decision makers no longer need to do this! They just need to respond myopically to the broadcast forecasts as if they were correct.
>
> **Regret and CCV bounds in the paper are only against benchmarks that are strictly-feasible**
>
> This is incorrect. We point the reviewer to Remark 4 where we mention the bounds that we get without any margin assumption. Note also that our algorithm is not instantiated by a margin parameter, so it simultaneously promises each downstream decision maker an infinite family of guarantees to margin-based benchmarks, parameterized by the margin, including a guarantee for the zero margin case.
>
> The reviewer is correct to point out that regret bounds with respect to a strongly feasible class is a stronger assumption than regret bounds to the full class assuming Slater's condition. However we here want to re-emphasize: 1. The key novelty of our work is that we study a more general problem than the standard setting (as our results apply to multiple decision makers), and 2. we give bounds that require no margin condition at all.
>
>
>
> **There does not appear to be computational benefits of the proposed agent decision rule/computation oracle over existing ones in the literature (e.g. unconstrained quadratic minimization).**
>
> We respectfully disagree. Our algorithm operates in a centralized manner --- generating predictions that can be consumed by multiple downstream decision makers. Existing methods in the COCO literature, however, would require each decision maker to run their own separate algorithm, instead of simply and straightforwardly responding to given predictions as we suggest in our work. In particular, as the number of downstream agents becomes larger, the computational benefit of our algorithm is even more pronounced.
>
> **The setting is restricted to the case of linear utility and constraint functions.**
>
> While this is true, our results are stronger than previous work in the COCO setting as we study a more general setting, and provide stronger dynamic benchmarks.
>
>
> **The paper mentioned that the regret notion it considers is stronger than the dynamic regret considered by prior work in COCO. However, it is unclear how this is the case.**
>
> Even in the standard COCO setting, we give qualitatively stronger bounds in the sense that we promise dynamic \emph{swap} regret --- the dynamic  analogue of swap regret, usually studied in relation to its (non-dynamic) marginal analogue external regret. Swap regret promises regret not just to the best fixed action in hindsight, but to the best \emph{remapping} of actions to actions using an arbitrary swap function. Similarly, we provide a guarantee to a slowly drifting \emph{sequence} of swap functions. As far as we are aware, bounds of this sort have not previously been considered in the COCO literature.  In particular, our proposed benchmark is stronger than the vanilla dynamic regret benchmark studied in Guo et al. (2022) or Liu et al. (2022). The order wise bound is given in Corollary 4 and roughly scales with $\max{(T^\frac{1}{4}\Delta(\vec\phi)^\frac{3}{4}, \sqrt{T\Delta(\vec\phi)})}$.

---

### Official Review · Reviewer_CAP7 · 2025-10-31

**Soundness:** 3
**Presentation:** 3
**Contribution:** 3
**Rating:** 6
**Confidence:** 4

**Summary:**

This paper extends the framework of online omniprediction with long-term constraints to obtain dynamic regret bounds. The authors propose an algorithm that provides regret and cumulative constraint violation guarantees on arbitrary subsequences, with dependence on the number of subsequences that is logarithmic rather than linear. This exponential improvement enables meaningful dynamic regret bounds which were not previously possible. Agents employ a stateless prediction-based decision rule contrary to prior works which need to maintain a notion of state. The paper shows $\tilde{O}(\sqrt{T})$  regret and constraint violation when a strictly feasible benchmark exists and $\tilde{O}(T^{2/3})$ bounds when feasibility is not strict.

**Strengths:**

The paper studies an interesting extension of the omniprediction problems and provides interesting technical solutions. I found the result for $\lambda=0$ quite surprising.

**Weaknesses:**

The paper claims that the standard benchmark in online problems with long term constraints is the best action satisfying the constraints in every round. However, several works instead compare against distributions over actions which are feasible on average (e.g., Castiglioni et al. (2022) which is already cited, and Immorlica et al. (2022)).

While the authors note that most literature assumes a single feasible action across all rounds, there are papers that relax this (e.g., Castiglioni et al. 2024). Clarifying how this work fits relative to those assumptions would improve positioning.

The introduction could expand paragraph 79 to explain why the omniprediction framework is important and provide a bit of additional context for readers not familiar with the problem.

The paper assumes the adversary selects a distribution over outcomes each round, from which the actual outcome is sampled. Is there any assumption on the distributions in the paper (e.g., not too peaked)? Otherwise what’s the advantage in describing the adversary in this way? It would be good to explicitly state that no smoothed assumptions are used it that’s the case.

I find a bit counterintuitive that the benchmarks for dynamic regret still requires per-round feasibility. While with static baselines having the constraint to hold for each t makes sense, here I would expect the constraint to hold over the entire time horizon. Since the learner’s constraint satisfaction is averaged over time, it seems more natural to allow the comparator sequence to be feasible on average as well. This is an important point that should receive further discussion.

Studying the case in which the feasibility parameter is set to 1/sqrt{T} feels somewhat arbitrary. What if we know there exists a strictly feasible solution but we don’t know the exact value of lambda? Can I still obtain some guarantees? This should be properly discussed.

Compared to standard online problems with long-term constraints, I found the guarantees in absence of Slater’s condition to be quite surprising. It would strengthen the paper to provide an intuitive explanation of what structural properties make this possible, especially since no comparable results exist in standard online learning with constraints. Right now this is only mentioned in one remark in the main paper.

Section 3 lacks an explicit high-level description of the algorithm (paragraph 343). Even a short overview would improve readability.

The introduction is five pages long, while some technical sections (notably 3.4) are dense and quite opaque. Condensing the intro and expanding the discussion of the main technical ideas would make the paper much more readable.

Minor comments

* In Definition 4, formally define “piecewise feasible” to prevent ambiguity.
* Both the main text and appendix contain several out-of-margin equations that need formatting adjustments.


Immorlica, N., Sankararaman, K., Schapire, R., & Slivkins, A. (2022). Adversarial bandits with knapsacks. Journal of the ACM, 69(6), 1-47.
Castiglioni, Celli, and Kroer. 2024. Online learning under budget and ROI constraints via weak adaptivity. In Proceedings of the 41st International Conference on Machine Learning (ICML'24)

**Questions:**

see above

---

> ### Author Response · Authors · 2025-11-21
> **Response to reviewer**
>
> Thank you very much for your close reading and your time. We are happy to hear you have found the results in the paper interesting. We have incorporated your suggestions regarding the presentation of the paper in the revision.
>
> Below are answers to specific questions:
>
> **Does the learner need to assume (or select) the value of lambda ahead of time**
>
> Great question. You ask why we analyze $\lambda = 1/\sqrt{T}$. This is just an example of an infinite family of guarantees that our algorithm simultaneously promises. This is because our algorithm is not parameterized by the margin parameter --- it is used only in the analysis. So the decision maker is guaranteed simultaniously the corresponding regret bound to each benchmark class, parametrized by each value of $\lambda$. This is inclusive of the bound we prove for the $\lambda = 0$ case. As we illustrate, we can give better bounds for bechmarks parameterized by larger values of $\lambda$ but there is no need to pick and choose --- we get them all.
>
> **Benchmarks --- best feasible action vs. best feasible distribution over actions**
>
> The reviewer is right that other works (including the ones cited) study variants of the learning with constraints problem corresponding to arbitrary convex action spaces, and compare to benchmarks that can represent the best distribution on base actions in hindsight. The problem we model each downstream decision maker as solving is an experts problem with a fixed set of discrete actions (not a general convex optimization problem), and our benchmark is defined with respect to their best fixed action (not distribution over actions) in hindsight. This is consistent with much of the prior work on omniprediction.
>
> **Assuming a single feasible action across all rounds**
>
> The benchmark consisting of the set of actions that are feasible across all rounds is widely considered to be the standard/basic benchmark in the literature on learning with long-term constraints.  The reviewer is right to point out that the work of Castiglioni et al. 2024 relaxes this assumption to instead require local feasibility (with margin). However, the cost that Castiglioni et al. 2024 pay for using this more relaxed benchmark is that their guarantees hold for either stochastic or oblivious adversaries, and provide only fractional optimality guarantees. We study the fully adaptive adversarial setting, and so the two kinds of guarantees are incomparable (even ignoring the omniprediction aspect that is the main novelty of our work). We have added  Castiglioni et al. 2024 in the related work discussion in our revision.
>
> **The adversary selecting a distribution over outcomes each round from which the realized outcome is sample as opposed to simply selecting a single outcome**
>
> You are right that our results do not require this generality --- the worst case adversary will be deterministic, and we allow this --- we place no smoothness (or other) assumptions on the adversary to restrict them. We state this more general setting only to emphasize that our results hold also in partially stochastic environments.
>
> **Benchmarks for dynamic regret  requiring per-round feasibility**
>
> The reviewer asks about our dynamic benchmark, which allows for changing actions but requires that the action at day $t$ satisfy the constraints at day $t$ (rather than just on average over all days). At this level of generality, the answer is yes. Note that the static benchmark is a special case of the dynamic benchmark. It has been known since the work of Mannor et al. 2009 (which we discuss on line 41) that even in the static case, competing with the best action that is ``marginally feasible'' is impossible against an adversary. This impossibility arising even in the static case is inherited by stronger dynamic benchmarks.
>
> We point the reviewer's attention to the fact that in both dynamic and static benchmarks, the constraint violation of the learner is averaged over time. The static benchmark is in fact a special case of the dynamic benchmark, where the action we wish to compete with is the same across all rounds. Given this perspective, it is perhaps not surprising that obtaining positive results against the more difficult dynamic benchmark requires at least the feasibility assumption made in the static benchmark case.
>
> Competing with the more permissive benchmark of actions that are feasible on average (or ``marginally'', rather than on every round) is known to be impossible, as we mention when citing the result of Mannor et al. 2009 (line 41).

---

### Official Review · Reviewer_Djoj · 2025-11-01

**Soundness:** 2
**Presentation:** 3
**Contribution:** 3
**Rating:** 4
**Confidence:** 3

**Summary:**

The paper studies online omniprediction with long term constraints, where, in each round, a learner predicts an outcome then multiple agents select their respective actions based on the prediction. Agents gain utilities from their actions based on actual outcomes. The learner aims to generate predictions such that agents choose actions that maximize worst-case utilities while minimizing worst-case constraint violation.

The paper proposes an algorithm building on the Unbiased-Prediction algorithm from literature: in each round, learner makes predictions that satisfy decision calibration and infeasibility calibration, and agents play constrained best response to the predictions. The authors prove that the algorithm has regret and constraint violation guarantees on arbitrary subsequences. Moreover, the provided regret guarantees are with respect to newly proposed regret definitions that are stronger than the conventional dynamic regret definition in online learning, so the paper’s regret guarantees imply stronger bounds on dynamic regret compared to literature.

**Strengths:**

1.	The paper introduces new regret benchmarks that expand the current scope of online omniprediction with long-term constrains.

2.	The paper accounts for practicality by considering a stateless decision rule for agents.

3.	Theoretical results on both constraint violation and regret performance are presented.

**Weaknesses:**

1.	Insufficient discussion of algorithm: The paper provides very limited discussion of the actual algorithm, which was only mentioned from line 343-350 in page 7. The paper says that the proposed algorithm builds upon Unbiased-Prediction algorithm from Noarov et al. to make predictions that attain decision calibration and infeasibility calibration, but the implementation details are not discussed. How to ensure decision calibration and infeasibility calibration? Compared with Noarov et al., what, if any, computational challenges are introduced by the two calibration constraints?

2.	Lack interpretation of theoretical results: The paper has many theoretical results that are quite dense, and all results are presented with no or little interpretation. I think the authors should spend some efforts providing “plain-language” type explanation for some of the results and their implications. For example, how to think about the bounds on different regret definitions in Section 3.4? When applying the proposed algorithm in practice, how can the derived theory guide learner?

**Questions:**

Please see my questions listed in weaknesses. In addition, I have a question about agent’s decision rules. How robust is the provided theoretical guarantees to agents’ decision rules? For example, if agents are unable to choose the actual constrained best response, but only able to choose a suboptimal action or choose an action randomly, can the paper’s analysis extend to conclude performance guarantees?

---

> ### Author Response · Authors · 2025-11-21
> **Response to reviewer**
>
> First, a sincere thank you for your time and effort in reviewing our work!
>
> Below are answers to specific questions:
>
> **Insufficient discussion of the algorithm**
>
> Thank you for pointing this out; for lack of space we had deferred a description of the algorithm to the appendix, but in the revision we have moved it to the main text. Noarov et al. give a basic primitive (similar in generality to a tool like multiplicative weights) for the problem of making predictions that are unbiased subject to an arbitrarily specified collection of events. We use their tool applied to a collection of events that we develop in this paper that we show specifically give strong guarantees for our problem of omniprediction with long term constraints. We have added text about what this looks like in substantially more detail that we hope demystify the algorithm for the reader.
>
> **Lack of interpretation of theoretical results**
>
> Following your recommendation, we added discussion and more intuition in the revision for both the algorithm and the theoretical results throughout the paper.
>
> **How to think about the bounds on different regret definitions in Section 3.4?**
>
> An intuitive way to interpret the results is the following --- ``traditional'' regret bounds measure how well the learner has performed against a fixed benchmark of a single action. Dynamic regret bounds are more powerful as they measure the performance of the learner against a potentially changing benchmark between rounds. The performance of changing benchmarks can, of course, surpass the performance of any single fixed action throughout the interaction hence the bounds we give are much stronger than `traditional'' ones.
>
> Additionally, using a changing benchmark is especially useful in  learning problems with long term constraints, as it is very much possible that a single action satisfying all of the constraints across rounds does not  exist!  Dynamic benchmarks only require per-round feasibility which is a much weaker requirement, as the action can differ across rounds of the benchmark sequence.
>
> Finally, in this paper we actually give an even stronger form of dynamic regret bound that we term ``swap-dynamic regret'', where our dynamically changing benchmark is not merely a sequence of actions, but a sequence of arbitrary swap functions mapping actions to actions (standard dynamic regret is the special case in which each swap function ignores its input and always outputs the same action). Swap regret is important in strategic settings, as individual decision makers with swap regret bounds cannot be manipulated by opponents in ways that agents with standard regret bounds can be. We note that competing with this benchmark is novel in the long-term-constraints literature even in the standard setting of a single decision maker (i.e. not only in our more general omniprediction setting).
>
> **When applying the proposed algorithm in practice, how can the derived theory guide learner?**
>
> Our proposed algorithm is a direct suggestion for what the learner should do in order to provide predictions that are useful for downstream decision makers. When the learner does this we also have simple, actionable guidance for the downstream decision makers: act as if the learner's predictions are correct, and optimize accordingly. Our theory promises that decision makers who act in accordance with this advice will obtain strong worst case dynamic bounds on both their reward and constraint violation.
>
> **How robust is the provided theoretical guarantees to agents’ decision rules? For example, if agents are unable to choose the actual constrained best response, but only able to choose a suboptimal action or choose an action randomly, can the paper’s analysis extend to conclude performance guarantees?**
>
> This is a good question; we focus our paper on downstream agents who best respond, but this is not necessary: the events with which our predictions must be unbiased with respect to only need to be tailored to the decision rule used by the downstream decision maker. We can do this no matter what that decision rule is, so long as it is known (or approximable) by us. Of course the final regret guarantees will degrade with how sub-optimal the decision makers response rule is compared to best response, as they must even with perfect forecasts.

---

### Author Response · Authors · 2025-11-21
**To all reviewers**

We thank all of the reviewers for their careful reading and thoughtful comments. We respond to individual reviewer questions below. Here we want to make a couple of global comments about the novelty of our work and its relationship to the existing learning with long term constraints literature.

The most important thing to emphasized is:

1. The learning with long term constraints literature focuses on single decision makers, and derives algorithms they can use that offer them various individual guarantees. Our paper studies a different, more difficult problem: how a single learner who offers ``predictions as a service'' can serve predictions that can simultaneously be consumed by many downstream decision makers, who each have different objectives and constraints, in simple ways that nevertheless offer them strong guarantees. In particular we want our decision makers to be stateless, and to simply be able to trust our predictions and  myopically solve  constrained optimization problems as if our predictions were correct. This is the core innovation of our setting beyond the standard problem of learning with long term constraints.

2. We remark that even when considering the special case of a single decision maker (which maps to the traditional setting of learning with long-term constraints), our work introduces and competes with stronger benchmarks than the ones previously studied. Namely, our introduced notion of swap-dynamic regret generalizes  dynamic benchmarks studied in this literature literature. This is the ``dynamic'' analogue of swap regret (in comparison to the more standard external or marginal regret), and has not been previously studied in the learning with long term constraints literature. Of course this contribution is secondary to our main contribution 1).

We also acknowledge that due to space constraints in our initial submission, the details of the actual algorithm were opaque (and deferred to the appendix). In the revision we have used our extra page of space to bring these details back to the main body.

---

### Meta-Review · Area_Chair_fS6Z · 2026-01-07

**Summary:**

- Most reviewers raised concerns about the presentation of the paper. The technical contributions, theoretical results, and even the setting itself, are unclear to most reviewers. The paper could benefit from a major revision. For example, as Reviewer  (CAP7) pointed out (who gave the highest score 6), the introduction is 5 page long, but the technical sections (section 3 and 4) are dense and opaque. Reviewer Djoj also find the paper lack discussions of the algorithms and the interpretation of the theoretical results.

- The reviewers also question the technical novelty of the proposed method, for example,  the comparison with Behaved et al 2025 in terms of the methodology and theoretical guarantee, and the comparison with Guo et al . 2022 or Liu et al 2022 in terms of the proposed variation notion.

- Reviewer 59re also pointed out the similarity between the proposed problem and the existing framework of constrained online convex optimization (COCO). The setting considered in this paper is more restrictive than COCO because it is restricted to linear utility and constraint functions.

**Reviewer Concerns:**

**Resolved concerns**

- All reviewers ask some clarification questions due to the confusing presentation of the paper, which are addressed successfully.

**Remaining concerns**
- The presentation of the paper needs significant improvement.
- The technical contribution of this paper is unclear due to its similarity to COCO and several existing papers.
- The robustness of the provided theoretical guarantees to agents' decision rules should be more carefully discussed.
- More discussions on the constraint satisfaction and the relevant assumptions should be provided.

**Reviewer Scores:**

- Reviewer Djoj (score 4) is not likely to change the score.
- Reviewer CAP7 (score 6) is not likely to change the score.
- Reviewer 59re (score 2)  may increase the score to 3.
- Reviewer ivPL (score 4) is not likely to change the score.

---

### Decision · Program_Chairs · 2026-01-26

Reject